# Cortical feedback and gating in odor discrimination and generalization

**Gaia Tavoni** [1,2,5☬] *, **David E. Chen Kersen** [1,3☬], **Vijay Balasubramanian** [1,2,3,4]

**1** Computational Neuroscience Initiative, University of Pennsylvania, Philadelphia, Pennsylvania, United States of America, **2** Department of Physics and Astronomy, University of Pennsylvania, Philadelphia, Pennsylvania, United States of America, **3** Department of Bioengineering, University of Pennsylvania, Philadelphia, Pennsylvania, United States of America, **4** Department of Neuroscience, University of Pennsylvania, Philadelphia, Pennsylvania, United States of America, **5** Department of Neuroscience, Washington University in St. Louis, St. Louis, Missouri, United States of America

☬ These authors contributed equally to this work.
* gaia.tavoni@wustl.edu

## Abstract

A central question in neuroscience is how context changes perception. In the olfactory system, for example, experiments show that task demands can drive divergence and convergence of cortical odor responses, likely underpinning olfactory discrimination and generalization. Here, we propose a simple statistical mechanism for this effect based on unstructured feedback from the central brain to the olfactory bulb, which represents the context associated with an odor, and sufficiently selective cortical gating of sensory inputs. Strikingly, the model predicts that both convergence and divergence of cortical odor patterns should increase when odors are initially more similar, an effect reported in recent experiments. The theory in turn predicts reversals of these trends following experimental manipulations and in neurological conditions that increase cortical excitability.

**Data Availability Statement:** Codes are available on GitHub https://github.com/dkersen/olfactory-bulb.

**Funding:** VB and DK were supported by the Simons Foundation (https://www.

## Author summary

Contextual information can powerfully influence the neural representation and perception of sensory stimuli. Here, we propose a mechanism, based on unstructured feedback from the central brain to the sensory periphery, by which similar and different contexts lead to characteristic trends in convergence and divergence of cortical odor responses that are critically dependent on threshold to firing of cortical cells. The analysis predicts specific deficits in context-driven olfactory perceptual discrimination in neurological conditions of high cortical excitability, such as Alzheimer's disease.

## Introduction

Contextual information, which we define as the environmental information salient to a sensory experience, has a powerful effect on perception across a range of sensory modalities [1–8]. In olfaction, experiments have demonstrated the influence of context and task demands on the

simonsfoundation.org/) through MMLS grant
400425. GT was supported by the Swartz
Foundation (http://www.theswartzfoundation.org/),
award# 575556. The funders had no role in study
design, data collection and analysis, decision to
publish, or preparation of the manuscript.

**Competing interests:** The authors have declared
that no competing interests exist.

neural representation of odors at different levels in the olfactory pathway. In the olfactory bulb
(OB), where odor information is first processed before passing to cortex, context-dependent
changes in both single-neuron and collective bulb activity have been observed during and fol-
lowing learning [9–18]. Context can also reshape representations of odors in the olfactory cor-
tex: when odors are associated with the same or different contexts, the corresponding cortical
activity undergoes pattern convergence (increased response similarity) or divergence
(decreased response similarity) respectively [19, 20]. The mechanisms underlying such con-
text-induced transformations are of great interest in sensory neuroscience.

The OB and cortex are notably coupled to one another. Mitral cells (MCs) and tufted cells
(TCs) from the bulb project to several higher brain regions [21, 22]. In particular, experiments
have highlighted that the piriform cortex (PC) is activated by convergent and synchronous
inputs from the bulb: coincident activation of several glomeruli within a short time window
[23] is required to induce spiking in cortical pyramidal neurons, a mechanism that is thought
to be important for decoding complex combinations of chemical features [24, 25]. In turn, the
bulb receives extensive feedback from multiple areas of the central brain, including the PC
[26–31]. Such feedback can arise directly as a response to odor input (*i.e.* in a traditional feed-
back loop), but may also independently encode the odor's context, such as associated sensory
or reward information [12, 32]. At the cellular level, this feedback predominantly targets gran-
ule cells (GCs) and other OB interneurons, enhancing or suppressing their activity [26, 33–
37], but may also directly excite the MC/TCs [30, 38] or otherwise alter MC/TC activity via
neuromodulatory factors such as acetylcholine, serotonin, and norepinephrine [39, 40].

Experiments have demonstrated that this cortical feedback plays a critical role in OB func-
tion. Activation of feedback pathways can decorrelate MC output [33], enhancing odor dis-
crimination. Neuromodulation of the OB can also alter odor discrimination and, importantly,
adjust the influence of contextual information on activity in the bulb and odor perception [12,
15, 41, 42]. Consequently, disruption of these pathways impairs both associative [32, 41, 43]
and discriminatory abilities [15, 37, 41, 44]. Theoretical and computational studies have also
explored possible mechanisms by which feedback may impose these effects. Models of top-
down, direct cortical feedback to the bulb which include plasticity in the feedback and in cor-
tex have been able to reproduce odor association with visual context [12] and differences in
cortical reorganization during passive vs. active learning [45]; demonstrate adaptation to spe-
cific olfactory environments and odor tasks when guided by neurogenesis [46]; and explain
differential responses to the same odor under separate contexts [43]. Other models have dem-
onstrated further effects of neuromodulation on OB activity, such as normalization of output
neuron response [47], increasing spike synchrony [15, 48], and general enhancement of odor
discrimination [44, 47].

The potential role of such plastic feedback in driving contextual changes in odor discrimi-
nation and generalization is complicated by the apparently random projection of MC/TCs to
the cortex [24, 25, 49–51]. In particular, any changes induced by plasticity in feedback to the
bulb would appear to become scrambled in cortex, necessitating further plasticity at the synap-
ses between MC/TCs and cortical cells, and within the piriform cortex, to produce targeted
effects. Indeed, this dispersion of information from the bulb may underpin the distributed
character of the cortical representation of odors, with different odorants activating unique, but
completely dispersed groups of cortical neurons [25]. Likewise, cortical feedback fibers are dis-
tributed diffusely over the OB without any discernible spatial segregation [27, 31]. Conse-
quently, it is difficult to predict how different cortical feedback patterns affect overall OB
output and subsequent odor representation in cortex. Thus, global reorganization of the OB
network through synaptic plasticity to reflect the odor environment and its associated contexts

may also require significant time. This raises the question of how animals are able to effectively learn in the short term under these constraints [52].

Here, we postulate that diffuse feedback signals, carrying unstructured representations of context, can nonetheless modulate odor responses in the OB, even *without* synaptic plasticity, to provide a rapid trigger for effectively entraining robust convergence or divergence of odor patterns in piriform cortex (PC) depending only on the relative similarity between feedback patterns rather than the pattern identities themselves. These changes, which can be stabilized by plasticity in the recurrent connectivity of cortex [53–55], can in turn underpin generalization and discrimination in sensory behavior (Fig 1A). To test this hypothesis, we constructed a statistical, analytically tractable model of the OB and its projections to the PC, which we later extended by incorporating an anatomically-faithful network of interactions between OB excitatory and inhibitory cell types along with a realistic distribution of projections from the OB to the PC. This model provided a number of advantages unavailable via experimental methods, namely 1) the interrogation of a large number of neurons; and 2) precise control of the distribution of odor and feedback inputs.

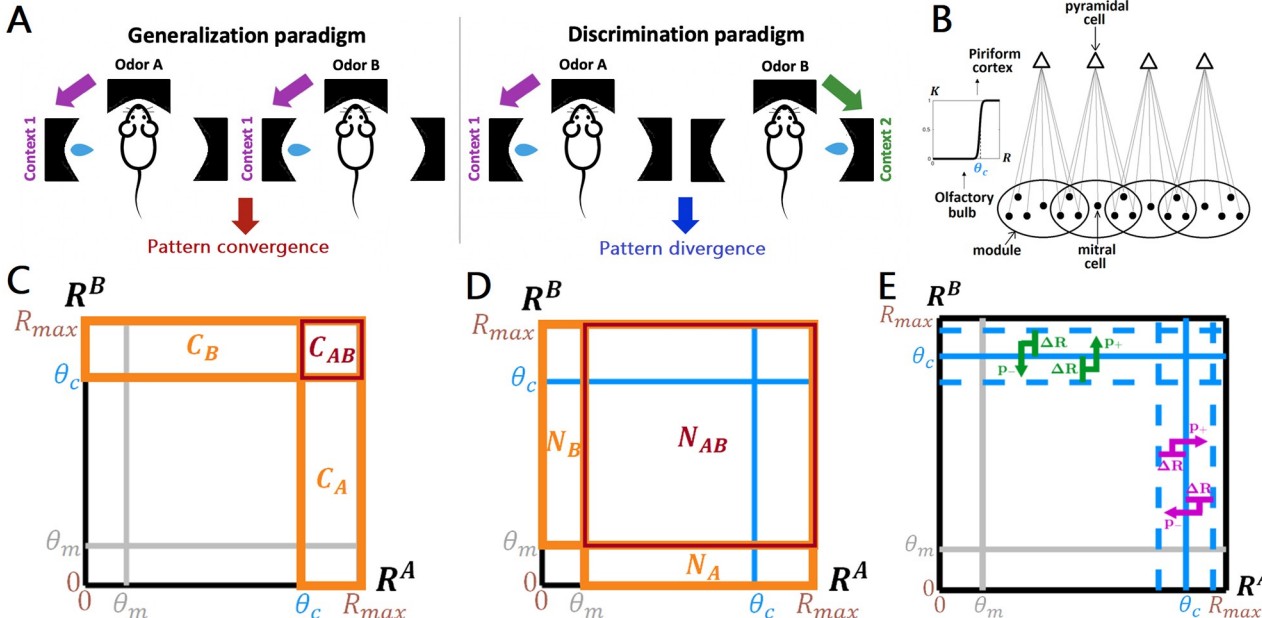

**Fig 1. A statistical mechanism for context-induced pattern convergence and divergence in the olfactory system.** (**A**) Context is information present in the environment that is salient to perception and behavior, e.g. the location of a reward associated with an odor. Left: in two-alternative forced-choice tasks, when distinct odors are associated with the same context (e.g. a left reward port), animals learn to generalize their choice (approaching the left port) across odors; after training, the neuronal ensemble responses to the odors in the PC are more correlated (pattern convergence, [19]). Right: when odors are associated with alternative contexts (e.g. opposite reward ports), animals learn to discriminate the stimuli and make different, odor-specific, choices (approaching the left versus right port); after training, the cortical ensemble responses are less correlated (pattern divergence, [19]). (**B**) In the statistical model, the olfactory bulb is represented as an ensemble of "modules" (black circles), with each module defined as the set of MCs (black dots) that project to the same cortical neuron (triangle), and the activation probability of cortical cells is a threshold function of the modules' responses. (**C**) A geometrical interpretation of Eq 5. The square represents the space of module responses to two odors (A and B). The number of cortical cells activated by odor A ($C_A$ in Eq (5)) is equal to the number of modules with responses higher than the cortical activation threshold $\theta_c$ (orange rectangle labeled $C_A$); likewise for $C_B$ (orange rectangle labeled $C_B$) and $C_{AB}$ (red rectangle). (**D**) $N_A$ (orange rectangle labeled $N_A$), $N_B$ (orange rectangle labeled $N_B$), and $N_{AB}$ (red rectangle) represent the number of modules responsive to odor A, odor B, or both odors, respectively. Because of the assumption of uniformity, $C_A$, $C_B$, and $C_{AB}$ can be re-expressed in terms of these values (Eq 1). (**E**) Stochastic feedback representing contextual changes in the distribution of module responses. Modules with responses to either odor within $\Delta R$ from the cortical activation threshold and targeted by positive/negative feedback (with probabilities $p_+$ and $p_-$, respectively) are brought above/below the cortical threshold, changing similarity between the cortical odor representations (Eq 7).

The model shows that changes in bulb firing rates driven by feedback lead to divergence or convergence of cortical odor representations. We had expected that, for odor discrimination, odors that were initially more similar would show greater divergence, while, for odor generalization, odors that were initially more similar would show less convergence, in order to achieve the same final degree of similarity. To our surprise, we found that, while the magnitude of pattern divergence indeed increased with initial odor similarity, so too did the extent of pattern convergence. These predictions of the model, which accord with results from recent experiments [19, 20, 56], follow from the general statistics of unstructured contextual feedback to the olfactory bulb, are compatible with the diffuse feedforward projections to cortex, and thus hold for realistic odor input distributions and feedback patterns. Notably, our results require that cortical units selectively gate coincident inputs, which PC pyramidal neurons are known to do [23, 25]. The model also predicts that increases in cortical excitability, either by experimental manipulation or in neurological conditions such as Alzheimer's disease [57, 58], will alter or even reverse these trends, and hence induce characteristic modifications of contextually-driven change in odor perception and behavior. Ultimately, our study provides a new approach to understanding the role of feedback in olfaction as well as the relationship between central and peripheral brain regions in sensation.

## Results

### A statistical mechanism for associating contexts to odors

Our model contains a sensory layer and a cortical layer, the olfactory bulb (OB) and the piriform cortex (PC) respectively, consistent with the basic anatomical features of the olfactory system (Fig 1B). The sensory (OB) layer consists of $N$ "modules", where each module represents the set of mitral cells (MCs) in the OB that project to the same pyramidal neuron in the cortical (PC) layer. Consequently, the cortical (PC) layer consists of $N$ units as well, with each unit representing a pyramidal neuron receiving direct inputs from its corresponding module in the sensory layer.

**Modeling odor inputs.**   We summarized the response of the $i^{\text{th}}$ OB module to an odor input by the change in the total firing $R_i$ of its constituent MCs compared to baseline, where $0 < R_i < R_{\max}$. Thus the response for the bulb as a whole was described by the module firing rate vector

$$\mathbf{R} = (R_1, R_2, \ldots, R_N) \tag{1}$$

These responses were then transformed into a cortical response vector $\mathbf{K}$ via an element-wise nonlinear activation function $f(\mathbf{R})$, such that:

$$\mathbf{K} = (K_1, K_2, \ldots, K_N) = (f(R_1), f(R_2), \ldots, f(R_N)) \tag{2}$$

**Modeling context-induced feedback inputs.**   We modeled context as the effect produced by cortical feedback inputs to the OB. In the OB, such feedback can lead to increases or decreases in MC firing and consequently in the strength of inputs to PC, represented in our model by the module firing rates. Cortical excitation of interneurons such as GCs and periglomerular cells reduces MC firing [26–31, 38], while other modes of feedback, such as direct excitation of MCs [30, 31], neuromodulation of MC excitability [42, 59–62], and excitation of deep short axon cells (which drives feedforward inhibition of GCs) [26, 33, 37] can enhance MC activity. As a result, we represented the effect of feedback as a change in module firing

rates:

$$\Delta \mathbf{R} = (\Delta R_1, \Delta R_2, \ldots, \Delta R_N) \tag{3}$$

Where $\Delta R_i < 0$ and $\Delta R_i > 0$ respectively represent decreases and increases in the overall firing rate of the $i^{\text{th}}$ module.

Additionally, we assumed that the similarity between changes in module responses induced by any two given feedback inputs, which we quantify using the cosine similarity between the respective $\Delta\mathbf{R}$s, reflects the correlation between the contexts that elicit those feedback inputs. (The cosine similarity between vectors $\mathbf{v}$ and $\mathbf{w}$ is defined as $\mathbf{v} \cdot \mathbf{w}/\|\mathbf{v}\| \, \|\mathbf{w}\|$ where $\cdot$ is the inner product and $\|\mathbf{v}\|$ is the norm of $\mathbf{v}$). In the most extreme case, feedback inputs that induce identical changes in the modules' responses represent identical contexts, e.g. two odors being associated with a reward at the same location (Fig 1A, left). Conversely, those that induce uncorrelated or anticorrelated changes represent different contexts, e.g. two odors being associated with a reward at different locations (Fig 1A, right).

**Defining cortical similarity.** The overlap $\rho_i$ between cortical responses to two odors A and B before feedback was quantified by the cosine similarity between the $\mathbf{K}$ vectors for the two odors. Then, the overlap $\rho_f$ between cortical responses for those two odors after feedback was simply the cosine similarity between the new $\mathbf{K}'$ vectors, where $\mathbf{K}' = f(\mathbf{R} + \Delta\mathbf{R})$.

## Change in similarity between odor representations following feedback varies linearly with the initial similarity

To capture the overall features of the system while maintaining analytical tractability, we made the following simplifications. Firstly, since experiments have indicated that piriform neurons only respond if the amount of input they receive within a short window is above a certain threshold [23, 24], we approximated $f(\mathbf{R})$ as a step function:

$$K_i = f(R_i) = \begin{cases} 0 & \text{if } R_i < \theta_c \\ 1 & \text{if } R_i \geq \theta_c \end{cases} \tag{4}$$

where $\theta_c$ quantifies the threshold for cortical activation. Thus, in our model, the $i^{\text{th}}$ cortical neuron receiving input from its module was said to be "active" (i.e. $K_i = 1$) only when $R_i \geq \theta_c$. Because $f(\mathbf{R})$ is a step function, we can express the cortical similarity as:

$$\rho = \frac{C_{AB}}{\sqrt{C_A C_B}} \tag{5}$$

where, before feedback, $C_{AB}$ is the number of PC neurons active for both odors, and $C_A$ and $C_B$ are the numbers of PC neurons active for odors A or B respectively (Fig 1C).

As a second approximation, we set the module firing rate distribution to be roughly uniform above and below a value $\theta_m$, which we denote the "response threshold". Modules with responses above this threshold are considered significantly "responsive" to that odor, but since $\theta_m \ll \theta_c$ [23, 24], a responsive module may not necessarily be cortically active, as described above. With this approximation of uniformity, we can interpret the problem geometrically (Fig 1D). If we define $N_{AB}$ as the number of modules responsive to both odors A and B, and $N_A$ and $N_B$ as the numbers of modules responsive to odor A or odor B respectively, then $C_{AB}$, $C_A$, and $C_B$ are just rescaled versions of $N_{AB}$, $N_A$, and $N_B$ respectively, and can be re-expressed

accordingly:

$$C_A \quad = \frac{R_{\max} - \theta_c}{R_{\max} - \theta_m} N_A \tag{6a}$$

$$C_B \quad = \frac{R_{\max} - \theta_c}{R_{\max} - \theta_m} N_B \tag{6b}$$

$$C_{AB} \quad = \left(\frac{R_{\max} - \theta_c}{R_{\max} - \theta_m}\right)^2 N_{AB} \tag{6c}$$

As a third approximation, for modules that receive non-zero feedback, we set the magnitude of the change in the module firing rate due to feedback to be the same for all modules. These approximations allowed us to derive analytical results. All our conclusions remain valid after relaxing these assumptions in numerical simulations of the model (S1 Fig, S1 Text).

General patterns of feedback, which vary in the number of modules they reach as well as in their ratio of excitation to inhibition, require an examination of many alternative cases. In the Methods, we carry out this examination and show analytically that for a pair of odors associated to specific contexts, each of which produces unstructured cortical feedback, the general relationship between the initial similarity $\rho_i$ of their cortical representations and the change in that similarity following feedback $\Delta\rho$ is linear:

$$\Delta\rho = Q_2 \, \rho_i + Q_1 \tag{7}$$

with $Q_1$ and $Q_2$ being complicated functions of the parameters of the feedback and the response thresholds defined above (full derivation in Methods). The linear relationship can be understood intuitively as follows: feedback changes odor similarity by pushing some modules above or below the cortical threshold (depending on the feedback sign); these activated/deactivated modules are typically close to the cortical threshold and thus odor-responsive before feedback (Fig 1E); thus, they can be expressed as a fraction of the number of odor-responsive modules. Since the response similarity $\rho_i$ can also be re-expressed in terms of the number of odor-responsive modules (Eqs 5 and 6), one component of $\Delta\rho$ will be proportional to $\rho_i$, hence the $Q_2 \, \rho_i$ term in Eq 7. If feedback is sufficiently strong compared to the difference between the cortical and response thresholds, it may also activate for both odors some modules that are not odor responsive. This effect gives rise to the constant term $Q_1$. Note that the effects we are reporting are not a simple result of similar and different feedback producing corresponding changes in firing rates. Indeed, because we are considering unstructured feedback to the olfactory bulb, which then projects to cortex, we will see that the effects can be partially *reversed* by increasing cortical excitability, an observation that may serve as a potential experimental test.

To illustrate the general results, we can examine special cases. For example, suppose all modules receive the same excitatory feedback of strength $\Delta R$ for both odors, so all modules with responses to odor A or B that are close enough to the cortical threshold ($\theta_c - \Delta R < R_i < \theta_c$) are brought above threshold by the feedback; as a result, the cortical neurons to which they project are activated by the feedback. These context-induced changes are equivalent to an effective decrease of the cortical activation threshold by $\Delta R$ for both odors (Fig 2A). Thus, by computing the similarity via Eq 5 before and after replacing $\theta_c$ with $\theta_c - \Delta R$ in Eq 6, the resulting change in similarity can be expressed:

$$\Delta\rho = \frac{\Delta R}{R_{\max} - \theta_c} \rho_i \tag{8}$$

which demonstrates pattern convergence that increases linearly in magnitude with initial odor similarity.

Similarly, we can consider a simple situation where all modules receive excitatory feedback for the first odor and inhibitory feedback for the second odor, both with uniform strength $\Delta R$. By an argument similar to the one above for the case of identical feedback, these effects are equivalent to an effective decrease and increase of $\theta_c$ by $\Delta R$ for the first and second odors respectively (Fig 2B). By then computing the similarity before and after replacing $\theta_c$ with $\theta_c - \Delta R$ for the first odor and $\theta_c + \Delta R$ for the second in Eq 6, the change in similarity can be

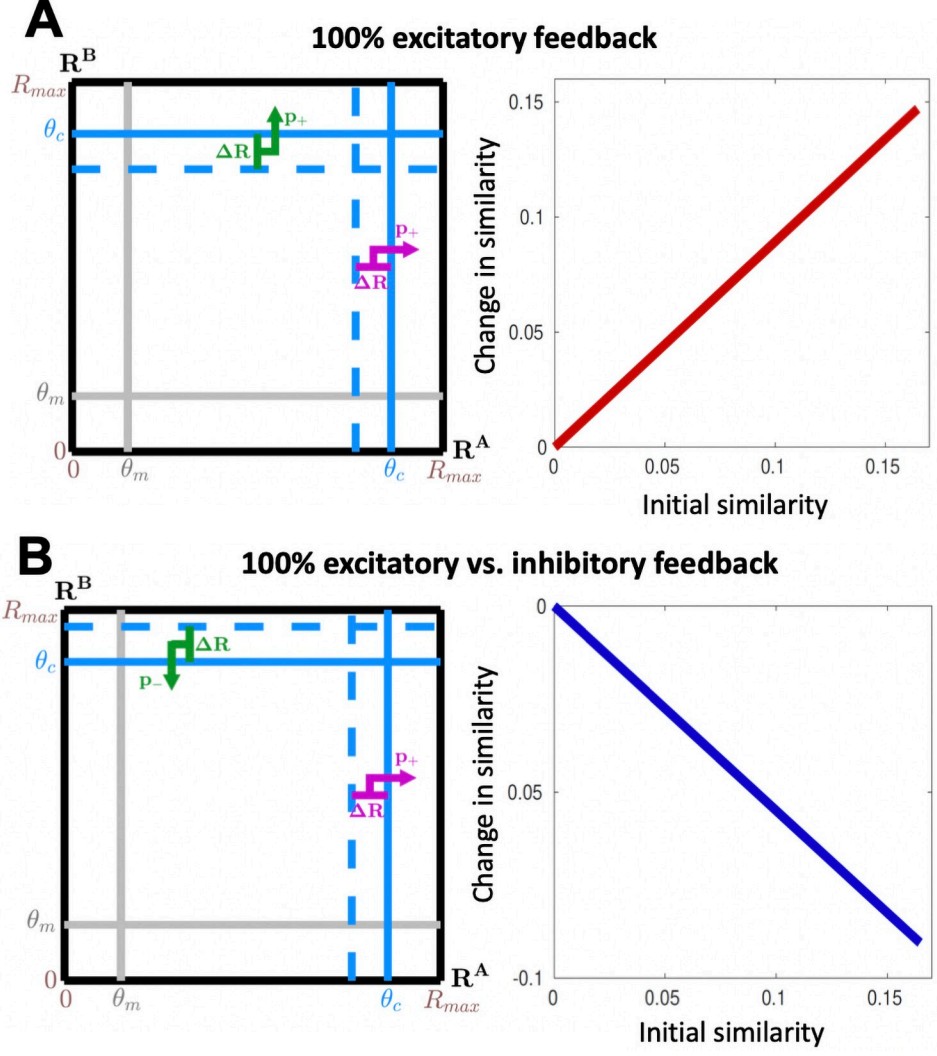

**Fig 2. Purely correlated or anticorrelated feedback induces proportionally increasing pattern convergence or divergence, respectively.** (**A**) Left: When feedback is identical and positive to all bulb modules for both odors, increasing their responses by $\Delta R$, its effect on the cortical activity is equivalent to a decrease of the cortical activation threshold by $\Delta R$. This results in a positive change in similarity between the cortical odor representations (i.e. pattern convergence). Right: The amplitude of this effect increases linearly with initial odor similarity. (**B**) Left: When the feedback is positive to all bulb modules for odor A and negative for odor B, the change in cortical activity is equivalent to that induced by a decrease/increase of the cortical activation threshold by $\Delta R$ for odor A/B respectively. This results in a decrease in similarity between cortical odor representations, i.e. pattern divergence. Right: The amplitude of this effect increases linearly with initial odor similarity.

expressed:

$$\Delta\rho = \left( \frac{\sqrt{(R_{\max} - \theta_c)^2 - \Delta R^2}}{R_{\max} - \theta_c} - 1 \right)\rho_i .$$

(9)

which demonstrates pattern divergence that again increases linearly in magnitude with initial odor similarity.

The general scenario, with different ratios of excitatory and inhibitory feedback, different numbers of modules affected by the feedback, and different correlations between feedback patterns has an intricate parameter dependence in the linear coefficients of Eq (7) (see Methods). To illustrate these general results, we computed the slope $Q_2$ under a general feedback pattern in which $\sim$50% of the modules were affected by feedback, with the majority of that feedback ($\sim$75%) being negative (i.e., suppressing firing rates), and whose feedback strength $\Delta R$ was $\sim$20% of the maximum firing rate $R_{\max}$. Fig 3A shows that under these conditions, when the cortical threshold $\theta_c$ was high (so that about 10% of the cortical units were activated by each odor, as is typical experimentally [25]), highly correlated contexts induced pattern convergence, while uncorrelated or anticorrelated contexts led to pattern divergence. Moreover, this pattern convergence and divergence both increased linearly with the degree of initial similarity in the odor responses, just as in the special cases described previously. These results held across: (i) different feedback strengths, and over larger ranges of cortical thresholds for stronger feedback, (ii) different fractions of modules targeted by feedback, (iii) different fractions of negative versus positive feedback, and (iv) different overlaps between the sets of modules targeted by feedback for the two odors (Fig 3B–3E) (although if the fraction of negative feedback was low, the transition between pattern divergence and pattern convergence could occur at an intermediate value of the feedback correlation even for low cortical thresholds (Fig 3D, left)). These results also persisted when the module firing rate distribution was Gaussian and the nonlinearity was changed from a step function to a sigmoid function (S1 Fig, S1 Text), but they changed dramatically if the cortical layer and its associated nonlinearity were removed (i.e. for module $i$, $f(R_i) = R_i$) (S2 Fig, S2 Text).

## Mechanistic model

Above, to facilitate mathematical analysis, we described activity in the OB and the effects of feedback in terms of independent firing rates (and changes in firing rates) of OB modules. However, anatomically, these modules comprise overlapping groups of MCs, and feedback targets individual MCs along with the GCs that inhibit them [26, 30]. Moreover, these MCs and GCs are highly interconnected in a network. Consequently, the precise effect that feedback has on MC output during combined odor driven and feedback activation is difficult to anticipate. To account for these considerations, we refined our approach with a detailed mechanistic model employing a biophysically realistic network of the OB (Fig 4A and 4B).

Briefly, to create this network, we modeled each cell in terms of its dendritic tree, since interactions between MCs and GCs are dendrodendritic [65–69]. Roughly mimicking the anatomy, MC trees were laid out in laminar discs and GC trees projected up in inverted cones [70–73]. These cells were embedded in a 3-dimensional space representing the layers of the OB. The x-y location of the center of a MC dendritic disc depended on the location of its respective glomerulus (which was in turn placed randomly) [74, 75], while its z-position depended on whether it was a Type I or Type II MC [70–72]. The GCs in our model were deep-type (since MCs primarily interact with deep GCs) [73], so while the x-y location of the vertex of a GC tree was random, the z-position of the vertex of its tree was confined to roughly

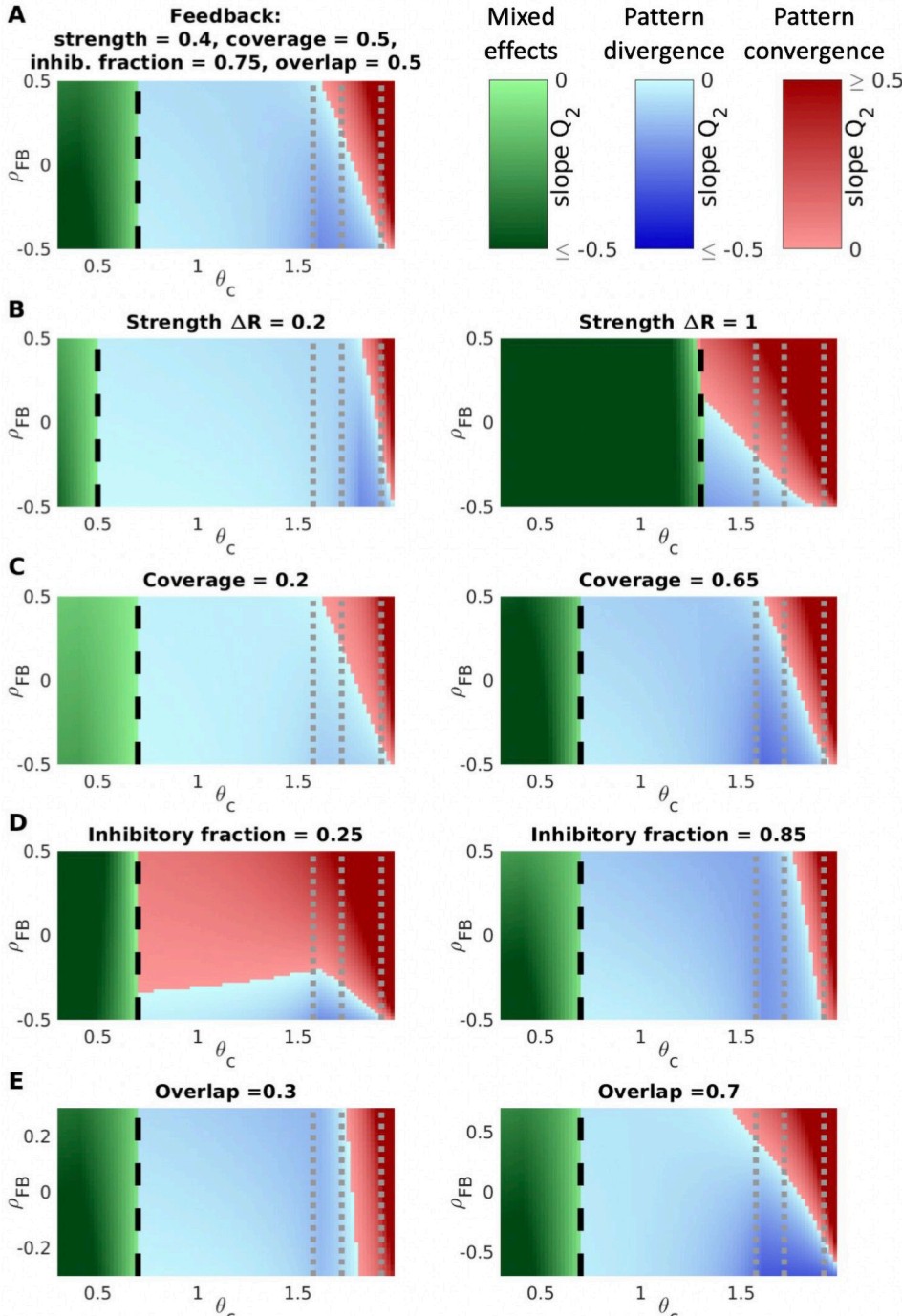

**Fig 3. Pattern convergence and divergence for general feedback.** (**A**) We computed the slope in Eq 7 as a function of the cortical threshold $\theta_c$ and the feedback similarity $\rho_{FB}$ for the following set of plausible feedback conditions: 1) feedback strength $\Delta R = R_{max}/5$; 2) feedback coverage $p_+ + p_- = 0.5$ for both odors; 3) fraction of feedback-affected modules that receive inhibitory feedback for one of the odors $p_-/(p_+ + p_-) = 0.75$ (the inhibitory fraction for the other odor varies with the feedback correlation along the y-axis); overlap between the subsets of modules affected by the feedback for the two odors $p_{both} = 0.5$. Pattern convergence and divergence increasing with initial odor similarity arise at values of the cortical threshold overlapping with the realistic range, with correlated and anticorrelated feedback, respectively. (**B–E**) Results generalize to a range of feedback conditions. Panels show effects of changing one feedback parameter, while maintaining others as in (A). Color maps represent values of slope coefficient $Q_2$ from Eq 7, with darker shades corresponding to higher $|Q_2|$, as a function of cortical activation threshold $\theta_c$ and feedback similarity $\rho_{FB}$.

**Red**: pattern convergence increasing with odor similarity ($Q_2 > 0$ and $Q_1 = 0$). **Blue**: pattern divergence increasing with odor similarity ($Q_2 < 0$ and $Q_1 = 0$). **Green**: mixed effects, i.e. pattern convergence decreasing with odor similarity and potentially turning into pattern divergence at high similarity ($Q_2 < 0$ and $Q_1 > 0$). **Gray dotted lines**: realistic range for the cortical threshold, activating 15%, 10% and 3% of cortex respectively in response to odor input [25]. **Black dashed line**: a threshold $\theta_c = \theta_m + \Delta R$ approximating the transition between pattern divergence/convergence and mixed effects. All panels: $\theta_m = 0.3$, $R_{max} = 2$, $p_{odor} = 0.6$ (probability that a given module has a significant response to an odor input).

the lower half of the model space. The probability of a connection between any given MC and GC was then a function of the geometric overlap between their dendritic trees (Fig 4A, details in Methods).

To simulate network dynamics, MCs and GCs were modeled as Izhikevich neurons [63], with parameters selected to match known electrophysiological data [72, 73, 76]. Odor input was represented as an oscillatory current into the MCs, while positive and negative feedback to the system were represented as pulses of constant excitatory current to a random subset of the

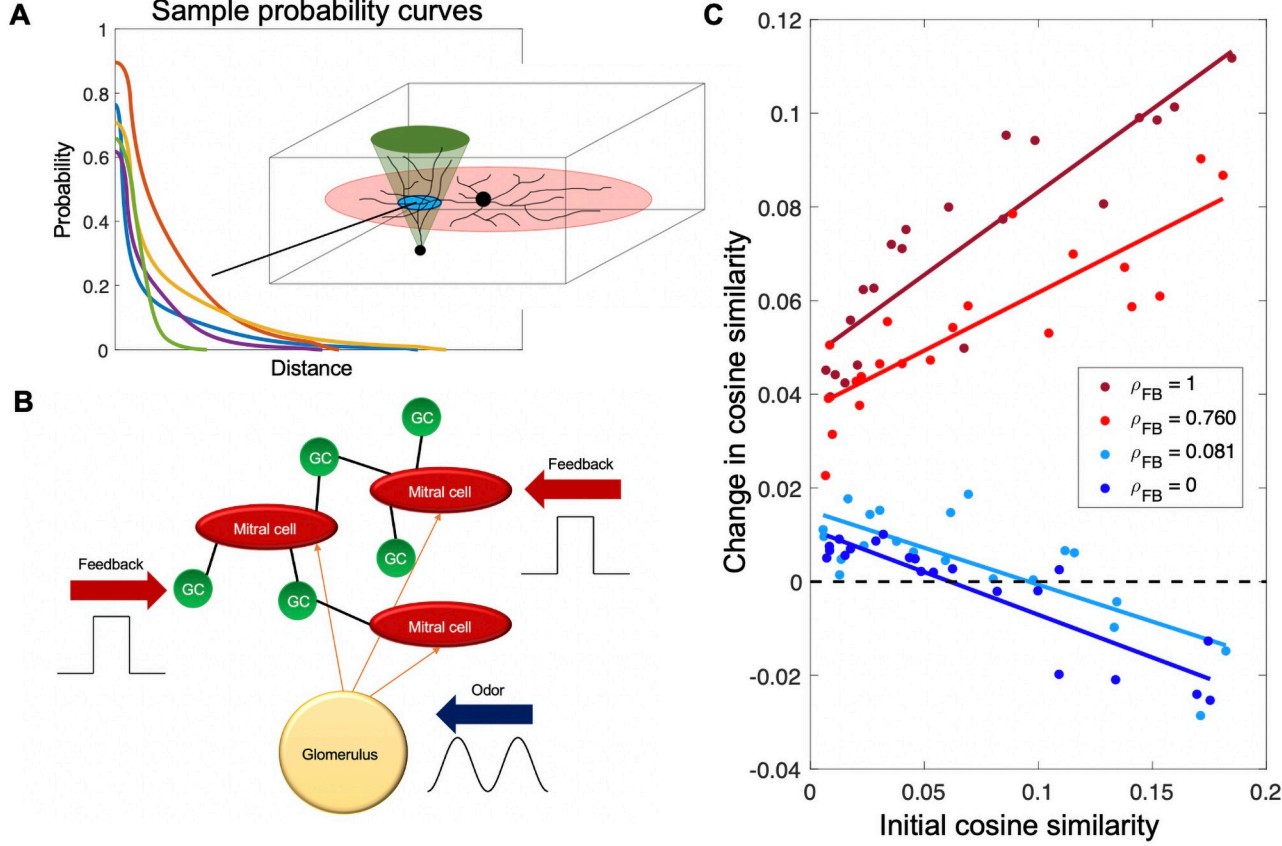

**Fig 4. A mechanistic model produces trends in pattern convergence and divergence similar to those predicted by the statistical framework.** (**A**) The probability of connection is a function of the overlap of the mitral cell and granule cell dendritic trees, here represented as a disc and cone, respectively. Different parameters in the mean-field distributions of these dendritic trees produce different probability curves for any given MC-GC pair. (**B**) The algorithm in (A) is used to generate a network of MCs interconnected by GCs. Network dynamics are simulated via the Izhikevich model [63, 64], with each neuron in the network receiving external inputs in the form of odor or feedback as well as reciprocal inputs resulting from the connectivity of the network. Odor input is modeled as an oscillatory current to mimic the respiratory cycle while external feedback is modeled via a constant current input pulse. (**C**) For high cortical thresholds, the cosine similarity between odor representations increases (pattern convergence) or decreases (pattern divergence) linearly in the initial similarity, according to the correlation in the feedback inputs. In (C), we simulated 10,000 MCs grouped into 500 glomeruli and 100,000 cortical cells, each sampling 7% of the MCs, with odor targeting 12% of the glomeruli, positive feedback targeting 8% of the MCs, and negative feedback targeting all MCs.

MCs and GCs respectively (Fig 4B). We determined that a sufficiently large number of MCs (>8000) was required to achieve statistically stable results (S3 Fig). The large size (especially given that there are around an order of magnitude more GCs than MCs [77]) meant that computational limitations precluded us from directly performing the spiking simulation for all the different odor and feedback tests. We circumvented this limitation by using the spiking model to extrapolate distributions of firing rates in MCs following odor input and reciprocal feedback from GCs, both with and without external feedback to MCs and GCs (see Methods, S4 Fig). This allowed us to simulate the detailed dynamics of larger numbers of MCs efficiently by summarizing the effects of the granule cell network in terms of the distribution of effects on MC firing.

To simulate cortical responses, we allowed $K$ cells to sample randomly from a fraction $q$ of the $M$ MCs, with $q \sim 0.07$, consistent with experimental measurements [49]. Each group of sampled MCs thus constituted a module in the terminology used above. MCs were also partitioned into $G$ glomerular, non-overlapping sets, each representing the set of MCs associated to the same glomerulus. We modeled the response to an odor as the evoked firing rate over a single sniff (a length of time sufficient for a rodent to distinguish between odors [78]) in a fraction $f_{odor}$ of the glomeruli. Thus, the MCs in $f_{odor}G$ glomeruli had a non-zero probability of having a firing rate greater than 0, while the remaining MCs had vanishing firing rates, after subtracting baseline responses. For the fraction of MCs that were selected to receive odor input, the firing rate for each MC was drawn from a distribution fitted to data from the detailed spiking model of the bulb (S4 Fig).

We determined the input to each cortical unit from the sum of the firing rates of MCs that projected to it. To account for the effects of cortical balancing [79], we subtracted the mean cortical input from the input to each unit, and then passed the result through a nonlinear activation function (a sigmoid). This ensured that the strongly activated cortical units tended to receive the highest rate, and hence most coincident, inputs. This sequence of steps yielded a vector of odor-induced firing rates over a sniff as the cortical representation of the odor. Finally, we modeled feedback as a vector of firing rate changes in a fraction $f_{FB}$ of randomly selected mitral cells. These changes could be induced either by direct feedback to the mitral cells [30, 38] or indirect inhibitory feedback through the granule cells [26, 28, 30, 31, 33, 80], all of which have the net effect of modifying MC firing rates. The specific pattern of feedback (*i.e.* the affected MCs and the net change in firing rate of each MC) effectively defined the context associated to the odor presentation.

**An *in silico* experiment on pattern convergence vs. pattern divergence.**   We tested how similarities and differences in feedback affected the cortical representation of odors in our mechanistic model. To this end, we generated pairs of odors, with each odor targeting $f_{odor}G$ glomeruli, where the two odors shared different fractions of targeted glomeruli. We then computed the cosine similarity $\rho$ between cortical responses to pairs of odors before and after addition of feedback for different feedback regimes.

We first considered the situation where the threshold of the sigmoidal activation function is high, leading to sparse firing in the cortex as seen in experiments [25, 81–83]. We presented pairs of odor inputs, which had varying degrees of initial similarity, and excitatory feedback to subsets of MCs and GCs, where the feedback patterns for each odor also varied in their similarity. We found that strongly correlated feedback led to pattern convergence (increased overlap in cortical responses), while uncorrelated feedback led to pattern divergence (decreased overlap in cortical responses) for higher initial odor similarity. Notably the pattern convergence and divergence both increased linearly with the initial overlap of cortical responses (Fig 4C, S6 Fig).

Additionally, simulations in the spiking network showed that inhibitory feedback patterns in the MCs arising indirectly via excitatory feedback to the GCs were necessarily diffuse and unstructured. Specifically, targeting non-overlapping sets of about the same number of GCs in the same network nonetheless produced highly correlated changes in MC firing rates; however, when we compared changes in MC firing rates following feedback between two networks with the same MC arrangement but a different spatial configuration of GCs, this correlation decreased [84]. Thus, in our mechanistic model, the identity of the MCs which are affected by inhibitory feedback through the GCs depends less on the particular GCs targeted, and more on the overall network connectivity within the bulb; additionally, the degree to which the firing rate of individual MCs is affected depends primarily on the fraction of GCs targeted and the strength of the feedback current, rather than the particular GCs targeted. In our reduced network then, under the assumption that the network remained the same for both odors and that feedback patterns were of roughly similar strength and extent, inhibitory feedback was necessarily highly correlated for both odor patterns. Interestingly, presentation of odors with highly correlated inhibitory feedback produced pattern divergence, unlike in the case with highly correlated MC feedback (S5 Fig, green line). Additionally, presentation of excitatory feedback for one odor and inhibitory feedback for the other produced strong pattern divergence (S5 Fig, purple line).

All told, these results recapitulated the predictions of the statistical model in a detailed mechanistic setting. One difference from the abstract analysis is that in this mechanistic model the only form of direct negative feedback goes through the GC network and is thus non-specific to particular MCs. As a result, strongly anticorrelated feedback is hard to achieve, but may be possible in the brain through targeted neuromodulatory effects or through feedback that suppresses GCs. In addition, because the MCs are embedded in a GC network, excitatory feedback to MCs necessarily induces some inhibitory feedback disynaptically through the granule cells. Thus, in this realistic mechanistic model there are constraints on the achievable forms of net excitatory and inhibitory feedback. Stronger feedback antisimilarity would further enhance pattern divergence.

## A prediction: Partial reversal of effects when cortical excitability is increased

In the normal brain, pyramidal cells in the PC have a high threshold for activation and respond only when there is coincident input from multiple MCs [23, 25, 85]. However, both experimental manipulation and neurological conditions such as Alzheimer's disease can cause increases in cortical excitability, or, equivalently, lower the cortical activation threshold [57, 58]. Our model makes striking predictions for these conditions that can be tested experimentally.

First, Fig 3 shows that pattern convergence only arises by our proposed mechanism within a range of cortical activation thresholds that are high, and these thresholds include the typical values expected for pyramidal neurons to achieve realistic levels of cortical activation [25]. Thus we predict that increased excitability in the PC (or, equivalently, reduced activation thresholds) will impair pattern convergence and thus the behavioral ability to generalize. Specifically, if cortical excitability increases moderately, our model predicts that any feedback inputs (anticorrelated as well as correlated) will induce a weak pattern divergence effect, increasing with initial odor similarity (light blue regions in between the black and gray dashed lines in Fig 3). If the increase in cortical excitability is sufficiently large, our model predicts more complex effects (green regions in Fig 3), as explained below. Qualitatively, in the high-threshold regime, only modules that already have a significant response to an odor input ($R_i >$

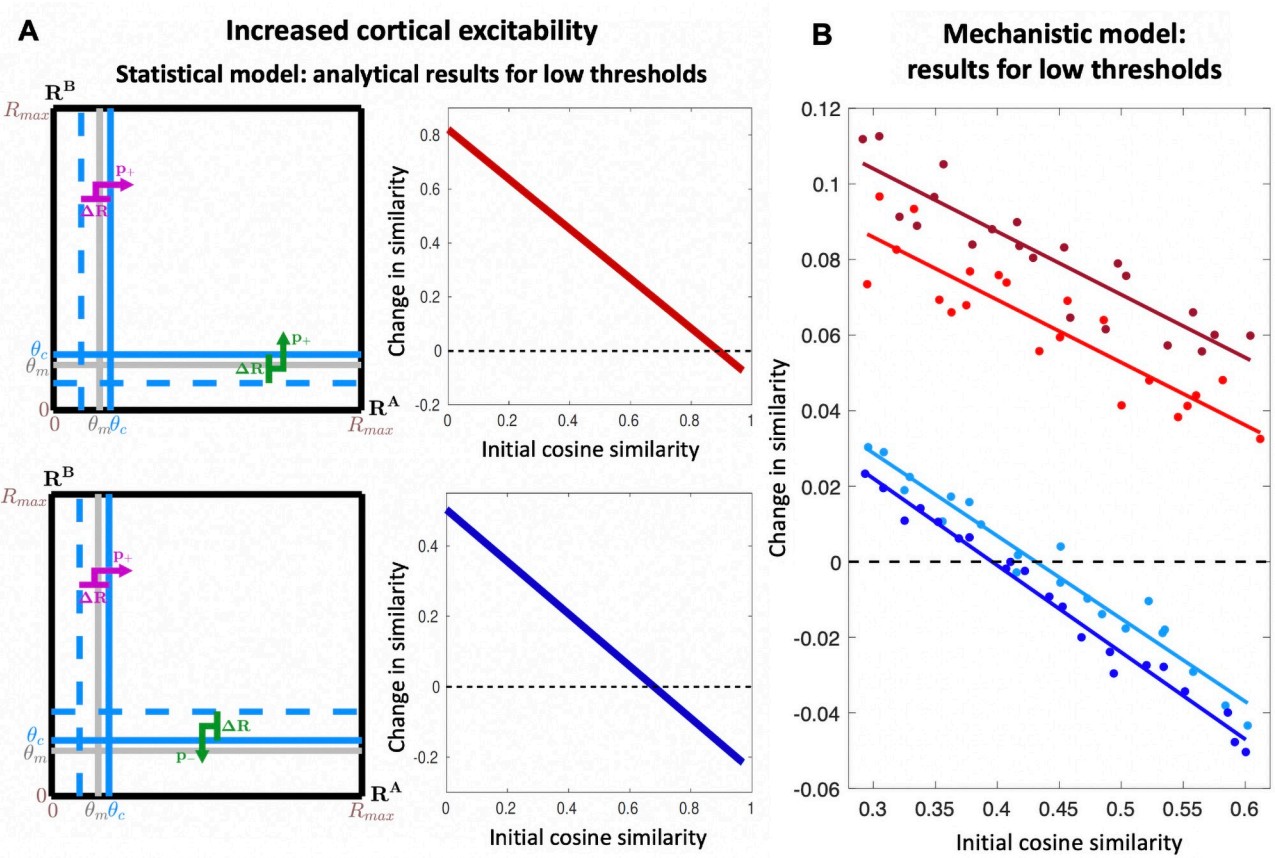

**Fig 5. The mechanism for pattern convergence and divergence requires a high-threshold transfer function.** Predicted trends in pattern convergence and divergence for increased cortical excitability, obtained from the statistical model and the mechanistic model with low activation thresholds. Low cortical activation thresholds yield effects that are qualitatively different from the realistic high-threshold case. (**A**) Analytical results obtained for $\theta_c = 0.35$ and the feedback scenarios of Fig 2A and 2B: red indicates 100% excitatory feedback for both odors; blue indicates 100% excitatory/inhibitory feedback for the first/second odor, respectively. Since $\theta_c$ is low, some modules that are not odor-responsive ($R < \theta_m$) are pushed by feedback above $\theta_c$ (sketches to left) changing similarity between cortical odor representations (right) differently from Fig 2A and 2B: correlated feedback inputs (red) yield pattern convergence decreasing with increasing odor similarity, and for high odor similarity, pattern convergence can turn into pattern divergence. Anticorrelated feedback inputs (blue) yield pattern convergence/divergence for low/high initial odor similarity, respectively. (**B**) Similar results are obtained from the mechanistic model (same simulation conditions as in Fig 4C).

$\theta_c - \Delta R > \theta_m$) can exceed the cortical threshold due to feedback, whereas if the threshold is sufficiently low ($\theta_c \lesssim \theta_m + \Delta R$), feedback can push some modules that were not initially odor responsive to be above the cortical threshold. This can influence the final cortical similarity and change results qualitatively (Fig 5).

Fig 5A shows analytical results from our statistical model at a low cortical threshold with identical (top) and opposite (bottom) feedback. We see that some of the effects at high threshold are reversed: (a) when the initial cortical similarity is high, similar contextual feedback to the bulb can lead to cortical pattern divergence instead of pattern convergence ($\Delta\rho$ becomes negative in Fig 5A, top right), (b) when the initial cortical similarity is low, dissimilar contextual feedback to the bulb can lead to cortical pattern convergence instead of pattern divergence ($\Delta\rho$ becomes positive in Fig 5A, bottom right), and (c) while pattern divergence for dissimilar contexts increases with increasing initial odor similarity (Fig 5A, bottom right), pattern convergence for similar contexts decreases with increasing odor similarity (negative slope in Fig 5A, top right), reversing the trend at high threshold. These reversals at low threshold are

confirmed by analysis of more general feedback conditions and response distributions (Fig 3, S2 Fig).

To confirm the predictions in the mechanistic model, we considered a situation where the threshold of the sigmoidal activation function is low, leading to greater cortical excitability, and hence broad activity in the cortical units. In this case we saw the same striking reversal: pattern convergence following strongly correlated feedback decreases with initial odor overlap (Fig 5B).

## Discussion

Context profoundly shapes odor perception [10, 14, 19, 45, 86–88], and previous studies have demonstrated the critical role of cortical feedback to the OB in the formation of odor-context associations [12, 32]. Cortical feedback can also enhance odor discrimination [33, 41, 42, 46, 47] and generalization [43]. Some studies have implicated feedback in the generation of beta oscillations, thought to be associated with olfactory learning [89–93]. Consequently, understanding the effect of cortical feedback on the OB is necessary to decode the broader relationship between context and perception in the olfactory system.

Our study contributes to this understanding by showing that diffuse feedback signals, carrying unstructured context representations, can modulate OB responses without synaptic plasticity to effectively entrain convergence and divergence of odor patterns in cortex despite the apparently random afferent projections from the OB. The model predicts that the resulting enhancement in generalization or discrimination should increase linearly with the initial odor similarity. This is especially surprising in the case of odor generalization, since we had expected that more dissimilar odors would show greater convergence in the presence of like contexts, in order to achieve an equal similarity in final cortical representation; in fact, initially similar odors converged more than dissimilar ones, suggesting that the cortex's "first impression" of similarity between a given odor pair plays a strong role in determining the influence of context. Moreover, our results also predict that these linear trends are critically dependent on the strong gating of the OB's projections to cortex. In fact, if the gating were weaker (i.e., if the threshold for cortical activation were lower, leading to less sparse activity), some of these effects would reverse. Our results are robust to broad changes in the statistics of odor inputs and feedback patterns, and are realized in a detailed mechanistic model of the circuit architecture of the olfactory bulb. We phrased our analysis in terms of responses to single odorants, but the model generalizes immediately to odor mixtures which are generally encountered by animals [94]. This is because we focus directly on the induced bulbar and cortical responses, rather than on the response of olfactory sensory neurons to specific odorants. Thus, the responses we study could equally well represent a single odor input or a mixture of odorants. Future work could extend our analysis all the way to the odor input by incorporating recent models of the nonlinear response of olfactory sensory neurons to mixtures of molecules [95, 96].

### Generality of the proposed mechanism across modalities

Our work focused on the olfactory system, as its known structure neatly delineates a simple circuit with few information-processing layers (the OB and the piriform cortex) and a clearly defined relationship between these layers, which can be individually adjusted and analysed in a model. However, similar mechanisms to the one we have proposed here could be implemented in other areas of the brain, and thereby support pattern convergence/divergence for other sensory modalities, especially given the importance of contextual information across the senses [1–8]. Indeed, all sensory cortices satisfy the key structural requirements of our statistical

model: (1) convergent projections from one layer to the next; (2) increasingly selective gating of the sensory inputs (as demonstrated by a reduction in neural activation levels across the sensory hierarchy [97]); and (3) feedback carrying information about the context of sensory stimuli towards the lower sensory layers. We showed that pattern convergence and divergence effects that emerge in neuronal networks with these features rely only on the statistics of feedback signals that are unrelated to the specific neuronal activation patterns induced by the sensory inputs: that is, feedback is not designed to target specific sensory neurons (e.g. based on their tuning properties), and does not require synaptic plasticity. Effective forms of pattern convergence/divergence arise statistically via a mechanism that simply adjusts the level of similarity in otherwise unstructured feedback patterns to reflect the similarity between contexts. Feedback signals with these simple properties can be implemented in many ways and in diverse sensory areas of the brain, making this mechanism applicable across modalities. Presuming this to be the case, the brain may also be able to forge relationships between different sensory systems simply by invoking randomly structured, but mutually correlated, feedback patterns to multiple sensory regions at the same time. This, for example, may facilitate strong correlations in the effects of context on the olfactory system and other sensory modalities including the gustatory system [98–102].

## Role of feedback to the olfactory bulb

Our results demonstrated that feedback to the OB could yield cortical pattern convergence or divergence varying linearly with the initial similarity between the two odor patterns. Because this effect is only dependent on the relationship between feedback patterns induced by different contexts, rather than on the activation of particular MCs or GCs in the bulb, the cortex has much more flexibility in producing pattern convergence and divergence via this mechanism. Moreover, this feedback will automatically induce a predictable level of pattern convergence or divergence, which can presumably be further adjusted by the cortex's native ability to mold odor representations in accordance with the animal's needs [53, 55, 103–105]. For example, piriform cortex could utilize feedback to induce broad initial changes to odor representation which are then solidified via the PC's inherent recurrent circuitry, which is necessary more generally to stabilize cortical odor representations [54]. Altogether, with these advantages, it is clear that the cortex, by targeting an upstream neural structure, can efficiently induce robust odor discrimination and generalization in its own odor representations, thereby suggesting one etiology for the extensive cortical feedback to the OB.

## Model predictions and experimental tests

Key behavioral predictions of our theory follow from the assumption that similarities and differences in the perception of an odor are directly related to the similarities and differences in their cortical representation. Thus, our theory suggests that perceptual learning should be strongly affected by cortical excitability, or, equivalently, the response threshold of cortical neurons. At low excitability (or high threshold), which is the normal state of sensory cortex, odors presented in related contexts should be perceived as more similar than they originally are, and the extent of this perceptual change should increase with the initial similarity. Therefore, the more similar the odors smell initially, the more similar they should seem after presentation in the same context. By contrast, if excitability is high (or threshold is low), the perceptual change in response to presentation in related contexts should be greatest for odors that were initially dissimilar. As a result, if the cortex of an animal performing a generalization task is highly excitable, stimuli that are very different should be disproportionately perceived as similar and potentially confused with one another, while perceptual association of initially

similar stimuli would be less effective or not achieved at all. Conversely, we predict that, at both low and high thresholds, the ability to discriminate odors associated with different contexts will increase with initial odor similarity.

Some of these predictions are already supported by experimental evidence from studies conducted in rodents. In particular, both [19] and [20] found that association of odor mixtures with different contexts decorrelated the cortical neural responses to the mixtures (pattern divergence) only in difficult discrimination tasks, i.e. when the odor mixtures were perceptually similar. This experimental finding is consistent with our prediction of a positive gain for pattern divergence as a function of the initial odor similarity. Furthermore, both studies reported an increase in cortical response similarity (pattern convergence) when odors were associated with the same context (the same reward port in [19], the same positive valence in [20]). In support of our model predictions, the perceptual effects measured in [19] also strongly suggest a positive gain for pattern convergence.

Our model also makes predictions about the different effects produced by different patterns of feedback. Interestingly, we show that the overall transition as a function of cortical threshold and feedback correlation from pattern divergence to convergence is generally invariant to changes in feedback parameters (Fig 3). Two additional statistical effects that our model identifies are that pattern divergence and pattern convergence can be achieved for low and high feedback correlation (1) over a larger range of cortical thresholds for stronger feedback (Fig 3B) and (2) even in the presence of relatively low cortical thresholds if the fraction of inhibitory to excitatory feedback is low (Fig 3D, left). Experimentally, these predictions could be tested by first engineering similar or different contexts for a pair of odors. Then, after training animals to associate each odor with its given context, one could measure the statistics of the consequent changes in OB and PC neural activity (e.g. via electrophysiology or calcium imaging), under the hypothesis that these changes reflect the influence of cortical feedback encoding context.

Overall, the predictions following from our proposed mechanism are readily falsifiable. Their rejection or further validation requires measurements of the similarity in cortical responses to odor pairs before and after odor discrimination/association training in two sets of conditions: a control set, in which the response properties of cortical neurons are not altered, and an experimental set, with heightened piriform cortical excitability (i.e. lower activation thresholds). An increase in excitability could be facilitated in a number of ways, for example via stimulation of other brain regions [106], or application of a GABA-A antagonist [107].

Our predictions could also be probed in patients showing neurological conditions of increased cortical excitability such as Alzheimer's disease [57, 58]. A clinical investigation could involve two participant groups, one with similar stage Alzheimer's disease and a healthy control. Both groups would experience a set of odors and rate the similarity of odor pairs. The participants would re-experience these odors with concurrent audio or visual stimulus as "context" (for example, monocolor images whose RGB values can be used to determine feedback similarity). Then they would be asked to again reassess the similarity between the different odor pairs. According to our model, we expect both the healthy controls and the disease group to report a decrease in similarity for initially similar odors that have been associated with different contexts, with the magnitude of this change increasing with initial odor similarity. However, the way in which each group perceives odors associated to similar contexts should differ markedly. In the case of the healthy group, odors that are initially perceived to be more similar should preferentially increase their similarity after presentation in the same context. Of course highly similar odors will have little room to increase their similarity before they are perceived to be identical. However, in the disease group, due to the decrease in cortical threshold associated with Alzheimer's disease, odors which are perceived to be initially most dissimilar should experience the greatest increase in perceived similarity following association to the same

context. To further explore our model's predictions in relation to neurodegeneration, we could also study how these trends in pattern convergence and divergence evolve with age. Under the same experimental paradigm, we expect the slope of the relationship between initial similarity and context-induced change in similarity to generally decrease with age, although this decrease may be nonlinear and any individual age bracket would demonstrate a variety of slope values (and hence phenotypes) [108].

## Methods

### The statistical model

**Parameters and assumptions.** We derive the results of the statistical model in terms of the parameters and assumptions in Table 1. Parameters $p_{\text{both}}$, $p_{\text{flip}}$, and the distribution of $\Delta R$ together control $\rho_{FB}$, defined as the cosine similarity in the response changes induced by feedbacks $F_A$ and $F_B$ for odors A and B. Assuming constant feedback strength $\Delta R$, $\rho_{FB}$ is simply:

$$\rho_{FB} = \frac{\Delta \mathbf{R}_A \cdot \Delta \mathbf{R}_B}{\| \Delta \mathbf{R}_A \| \| \Delta \mathbf{R}_B \|} = p_{\text{both}} - 2p_{\text{flip}} \tag{10}$$

where $p_{\text{both}}$ is the probability that a module receives feedback for odor B given that it receives feedback for odor A, and $p_{\text{flip}}$ is the probability that the feedback for odor B increases/decreases a module's firing rate given that the feedback for odor A decreases/increases it, respectively. The probability $p_{\pm}^B$, that feedback $F_B$ increases/decreases the firing rate of a module, can be related to $p_{\pm}^A$, the probability that feedback $F_A$ increases/decreases the firing rate, in terms of $p_{\text{both}}$ and $p_{\text{flip}}$:

$$p_{+}^B = \frac{p_{\text{same}} p_{+}^A + p_{\text{flip}} p_{-}^A}{p_{\text{both}}} \qquad\qquad p_{-}^B = \frac{p_{\text{same}} p_{-}^A + p_{\text{flip}} p_{+}^A}{p_{\text{both}}} \tag{11}$$

with $p_{\text{same}} = p_{\text{both}} - p_{\text{flip}}$. Similarly we can use Bayes' rule $\Pr(F_A < 0 | F_B > 0)\Pr(F_B > 0) = \Pr(F_B > 0 | F_A < 0)\Pr(F_A < 0)$ written as $p_{\text{flip}}^A p_{+}^B = p_{\text{flip}} p_{-}^A$ to write the probability that the feedback for A is negative given that the feedback for B is positive:

$$p_{\text{flip}}^A = \frac{p_{\text{flip}} p_{-}^A}{p_{+}^B} \tag{12}$$

We will use the expressions in (11) and (12) in our analyses.

**Analytical calculation of the cortical similarity.** The assumption of a uniform response distribution (see Table 1 and Fig 1C and 1D) allows us to write the number of modules responding over the cortical threshold $\theta_c$ as a proportion of the number of modules that are odor responsive (response higher than $\theta_m$):

$$C_A = \frac{\Delta R_c}{\Delta R_m} N_A \qquad\qquad C_B = \frac{\Delta R_c}{\Delta R_m} N_B \qquad\qquad C_{AB} = \left(\frac{\Delta R_c}{\Delta R_m}\right)^2 N_{AB} \tag{13}$$

Hence, the initial similarity between the cortical responses to odors A and B is a fraction of the initial similarity in the odor-responsive OB modules (Eq (5)):

$$\rho_i = \frac{C_{AB}}{\sqrt{C_A C_B}} = Q_0 \frac{N_{AB}}{\sqrt{N_A N_B}} \quad \text{with} \quad Q_0 = \frac{\Delta R_c}{\Delta R_m} \tag{14}$$

**Table 1. Parameters and assumptions of the statistical model.**

| | Parameters |
|---|---|
| **Thresholds** | |
| $\theta_m$ | module response threshold |
| $\theta_c$ | cortical response threshold ($\theta_c > \theta_m$) |
| $\Delta\theta = \theta_c - \theta_m$ | difference in thresholds |
| **Module responses** | |
| $R_{max}$ | maximum module response |
| $\Delta R_c = R_{max} - \theta_c$ | range of responses above the cortical threshold |
| $\Delta R_m = R_{max} - \theta_m$ | range of responses above the response threshold |
| $N$ | total number of modules |
| $N_\mu \ (\mu = A, B, AB)$ | number of modules with response greater than $\theta_m$ for odor A, odor B, or both ($\mu = AB$) |
| $C_\mu \ (\mu = A, B, AB)$ | number of modules above the cortical threshold $\theta_c$ for odor A, odor B, or both ($\mu = AB$) |
| $\Delta R$ | magnitude of change in firing rate for modules affected by feedback |
| **Module densities in response space (derived in terms of the parameters above)** | |
| $\rho(\mu) = N_\mu / \Delta R_m \ (\mu = A, B)$ | density of modules (modules/response range) responsive to odor A or B |
| $\rho(\bar{\mu}) = (N - N_\mu)/\theta_m \ (\mu = A, B)$ | density of modules (modules/response range) *not* responsive to odor A or to B |
| $\rho(AB) = N_{AB}/(\Delta R_m)^2$ | density of modules in response space (modules/area in response space) that are responsive to *both* A and B |
| $\rho(\bar{A}\bar{B}) = (N - N_A - N_B + N_{AB})/\theta_m^2$ | density of modules in response space (modules/area in response space) that are responsive to *neither* A and B |
| $\rho(A\bar{B}) = (N_A - N_{AB})/(\Delta R_m \times \theta_m)$ | density of modules in response space (modules/area in response space) that are responsive to A but *not* to B |
| $\rho(\bar{A}B) = (N_B - N_{AB})/(\Delta R_m \times \theta_m)$ | density of modules in response space (modules/area in response space) that are responsive to B but *not* to A |
| **Feedback** | |
| $F_\mu \ (\mu = A, B)$ | feedback associated to odor A or B |
| $p_\pm^\mu \ (\mu = A, B)$ | fraction of modules that increase (subscript +) or decrease (subscript −) their response due to feedback $F_\mu$ |
| $p_{\text{both}}$ | conditional probability that a module is affected by feedback for B given that it is also affected by feedback for A |
| $p_{\text{flip}}$ | conditional probability that the feedback for B increases/decreases a module's firing rate given that the feedback for A decreases/increases it, respectively |
| $p_{\text{same}} = p_{\text{both}} - p_{\text{flip}}$ | conditional probability that the feedback for B increases/decreases a module's firing rate given that the feedback for A also increases/decreases it |
| | **Assumptions** |
| response distribution (analytical model) | marginal and joint distributions of responses to odors A and B are separately uniform below and above $\theta_m$ |
| supplemental numerical analysis in S1 Fig | odor-response distribution and response change after feedback ($\Delta R$) are taken to be Gaussian |
| feedback coverage | both feedbacks target the same number of modules $p_+^A + p_-^A = p_+^B + p_-^B$ |
| feedback independence 1 | the probability that one feedback increases/decreases a module's firing rate is independent of whether the other feedback affects the module |
| feedback independence 2 | the probability that one feedback has no effect on a module given that the second feedback does affect it is the same regardless of whether the latter increases or decreases the firing rate |

We want to derive an analogous expression for the similarity between cortical responses after receiving feedbacks $F_{A,B}$ associated to odors A and B (Fig 1E). To do so we have to ask how $F_A$ and $F_B$ affect the numbers of modules $C_A$, $C_B$ and $C_{AB}$ that produce cortical responses to odors A, B, and both.

**Feedback-driven changes in modules responding to odor A or to odor B.** Let us start with the $C_A$ modules producing cortical responses to A. (The modules responding to B can be treated similarly). We have to consider two cases: (i) modules whose responses increase due to feedback, and (ii) modules whose responses decrease due to feedback. $C_A$ is increased by modules whose responses increase if their initial firing rate $R$ is less than, but within $\Delta R$ of, the cortical threshold ($\theta_c - \Delta R \leq R \leq \theta_c$). $C_A$ is reduced by modules whose responses decrease if their initial firing rate $R$ is greater than, but within $\Delta R$ of, the cortical threshold ($\theta_c + \Delta R \geq R \geq \theta_c$). For simplicity, we have assumed an initially uniform distributions of the $N_A$ module responses above $\theta_m$, and of the $N - N_A$ module responses below $\theta_m$. This means that we can write the change in the number of cortically responsive modules in terms of the proportions of modules within the two ranges indicated above. If the change in the module firing rates after feedback $\Delta R$ is smaller than the difference between the cortical activation and module response thresholds $\Delta\theta$, then only the $N_A$ modules with initial firing rate greater than $\theta_m$ matter for the analysis. This gives the change in the number of modules with cortical responses to A as:

$$\delta C_A = [p_+^A \min(\Delta R, \Delta\theta) - p_-^A \min(\Delta R, \Delta R_c)]\, \rho(A) \tag{15}$$

Here $p_\pm^A$ is the probability that a given module increases or decreases its firing rate, $\rho_A = N_A/(R_{\max} - \theta_m)$ is the density of responsive modules (modules/firing rate range), $\min(\Delta R, \Delta\theta)$ is the size of the interval of rates that can contribute to increasing $C_A$, and $\min(\Delta R, R_{\max} - \theta_c)$ is the size of the interval of rates that can contribute to decreasing $C_A$. If $\Delta R > \Delta\theta$, some modules that were previously unresponsive (below threshold $\theta_m$) can contribute to the cortical response after feedback, and we have to add this contribution to get:

$$\begin{aligned} \delta C_A^{\text{total}} &= \delta C_A + p_+^A \min(\Delta R - \Delta\theta, \theta_m)\, \rho(\bar{A})\, H(\Delta R - \Delta\theta) \\ H(x) &= 0 \text{ if } x < 0 \text{ and equals 1 otherwise} \end{aligned} \tag{16}$$

where $\rho(\bar{A}) = \frac{(N - N_A)}{\theta_m}$ is the density of modules below the response threshold, and $\min(\Delta R - \Delta\theta, \theta_m) = \min(\theta_m - (\theta_c - \Delta R), \theta_m)$ is the size of the interval below $\theta_m$ from which modules could become cortically activated after feedback. We included the step function $H(\Delta R - \Delta\theta)$ to indicate that the second term is only present if the effect of feedback is larger than the difference between thresholds. Through similar reasoning

$$\delta C_B^{\text{total}} = \delta C_B + p_+^B \min(\Delta R - \Delta\theta, \theta_m)\, \rho(\bar{B})\, H(\Delta R - \Delta\theta) \tag{17}$$

is the change in cortical responses to B after feedback.

**Feedback-driven changes in modules responding to both odors.** A module may increase or decrease its firing rate after feedback following one or both odors, and this feedback may be correlated. To calculate the resulting change in the number of modules reaching the cortical threshold for both odors, $\delta C_{AB}$, we must consider different possibilities for the initial firing rates as in the analysis of responses to single odors above. For example, if the response to odor A is just below the cortical threshold $\theta_c$, while odor B lies just above, then positive feedback after A combined with the absence of negative feedback after B triggers a cortical response for both odors. By contrast, if the odor B response lies just below $\theta_c$ and the response to A lies just above, then the pattern of feedback must be reversed to get cortical responses after both odors. Reasoning in this manner we can distinguish the 15 cases enumerated in Table 2.

**Table 2. All fifteen cases for feedback-driven changes in modules responding to both odors.**

| | | | | | $\delta C_{AB}^{total} = \sum_{i=1}^{15} \delta C_{AB}^{(i)}$ |
|---|---|---|---|---|---|
| (i) | $(\theta_c - \Delta R, \theta_m)$ | $(\theta_m, \theta_c)$ | $(\theta_c, \theta_c + \Delta R)$ | $(\theta_c + \Delta R, R_{max})$ | Contribution $\delta C^{(i)}$ to $\delta C_{AB}$ |
| 1 | | A,B | | | $p_+^A p_{same}[\min(\Delta R, \Delta\theta)]^2 \rho(AB)$ |
| 2 | | A | B | | $p_+^A\left(1 - p_{flip}\right)[\min(\Delta R, \Delta\theta)\min(\Delta R, \Delta R_c)]\rho(AB)$ |
| 3 | | B | A | | $p_+^B\left(1 - p_{flip}^A\right)[\min(\Delta R, \Delta\theta)\min(\Delta R, \Delta R_c)]\rho(AB)$ |
| 4 | | A | | B | $p_+^A[\min(\Delta R, \Delta\theta)]\max(0, \Delta R_c - \Delta R)\rho(AB)$ |
| 5 | | B | | A | $p_+^B[\min(\Delta R, \Delta\theta)]\max(0, \Delta R_c - \Delta R)\rho(AB)$ |
| 6 | | | A,B | | $-\left(p_-^A + p_-^B - p_-^A p_{same}\right)[\min(\Delta R, \Delta R_c)]^2 \rho(AB)$ |
| 7 | | | A | B | $-p_-^A[\min(\Delta R, \Delta R_c)]\max(0, \Delta R_c - \Delta R)\rho(AB)$ |
| 8 | | | B | A | $-p_-^B[\min(\Delta R, \Delta R_c)]\max(0, \Delta R_c - \Delta R)\rho(AB)$ |
| 9 | A,B | | | | $p_+^A p_{same}[\min(\Delta R - \Delta\theta, \theta_m)]^2 \rho(\bar A \bar B)H(\Delta R - \Delta\theta)$ |
| 10 | A | B | | | $p_+^A p_{same}[\min(\Delta R - \Delta\theta, \theta_m) \times \Delta\theta]\rho(\bar A B)H(\Delta R - \Delta\theta)$ |
| 11 | B | A | | | $p_+^A p_{same}[\min(\Delta R - \Delta\theta, \theta_m) \times \Delta\theta]\rho(A\bar B)H(\Delta R - \Delta\theta)$ |
| 12 | A | | B | | $p_+^A\left(1 - p_{flip}\right)[\min(\Delta R - \Delta\theta, \theta_m)\min(\Delta R, \Delta R_c)]\rho(\bar A B)H(\Delta R - \Delta\theta)$ |
| 13 | B | | A | | $p_+^B(1 - p_{flip}^A)[\min(\Delta R - \Delta\theta, \theta_m)\min(\Delta R, \Delta R_c)]\rho(A\bar B)H(\Delta R - \Delta\theta)$ |
| 14 | A | | | B | $p_+^A[\min(\Delta R - \Delta\theta, \theta_m)]\max(0, \Delta R_c - \Delta R)\rho(\bar A B)H(\Delta R - \Delta\theta)$ |
| 15 | B | | | A | $p_+^B[\min(\Delta R - \Delta\theta, \theta_m)]\max(0, \Delta R_c - \Delta R)\rho(A\bar B)H(\Delta R - \Delta\theta)$ |

The first four columns are indexed by ranges for the initial response rates of the modules before feedback. For example, $(\theta_m, \theta_c)$ indicates that the initial response lies between the response threshold $\theta_m$ and the cortical threshold $\theta_c$. Likewise $(\theta_c - \Delta R, \theta_m)$ indicates that the initial response lies below the response threshold $\theta_m$ but is sufficiently high that a feedback-driven addition of $\Delta R$ to the response rate would push the module above the cortical threshold. This case is only possible if the cortical threshold is sufficiently low and the strength of the feedback is sufficiently high. The letters in each row indicate the range in which the response lies for each of the two odors. The results in rows 9–15 include the step function $H(x)$ which is defined as $H(x) = 0$ if $x < 0$ and $H(x) = 1$ if $x \geq 0$.

The first four columns in Table 2 list the various relevant ranges for the initial firing rates of a module responding to odors A and B in which increase or decrease of firing by an amount $\Delta R$ after feedback can move a module above or below the cortical threshold $\theta_c$. The results for the contribution to $\delta C_{AB}$ presented in the last column all have the same general form:

$$\Pr(module\ increases/decreases\ response\ for\ one\ odor)\times$$
$$\Pr(same\ or\ opposite\ for\ the\ other\ odor)\times$$
$$(Area\ of\ the\ region\ in\ the\ response\ space)\times$$
$$(Density\ of\ modules\ in\ this\ region).$$

The results are written in terms of the probability that both feedbacks affect the module ($p_{both}$), the probability that the two feedbacks have same sign effects ($p_{same}$), and the probability that the two feedbacks have opposite sign effects ($p_{flip}$). See Table 1 for definitions of other parameters.

For example, in Row 1 of Table 2, we start initially with a module whose responses to both odors lie above the response threshold and below the cortical threshold ($\theta_m \leq R \leq \theta_c$). We then multiply the probability that feedback causes the response to both odors to increase ($p_+^A \times p_{same}$) with the number of modules that lie close enough to the cortical threshold such that feedback will take them over this threshold. This latter is evaluated by multiplying by the density of modules in the joint A-B response space ($\rho(AB) = N_{AB}/(\Delta R_m)^2$) (modules per unit

response space area) times the area of the region that lies close enough to the cortical threshold to cross over after feedback. Since the feedback is taken to have magnitude $\Delta R$, the latter area is the square of the minimum of $\Delta R$ and the interval between the thresholds $\Delta\theta = \theta_c - \theta_m$.

In Row 2 of Table 2 we consider a module which initially responds below the cortical threshold to odor A, but above the threshold for odor B. Thus, before feedback, this module does not contribute to $C_{AB}$, the count of modules with cortical responses to both odors. After feedback the module will contribute to this count if the response to odor A increases, and the response to odor B does not decrease. The probability of this happening is $p_+^A (1 - p_{\text{flip}})$, accounting for the first two factors in the result. To count the number of modules that experience this situation, as with the Row 1 result, we have to multiply by the density of modules in response space by the area in response space which lies close enough to the cortical threshold. This accounts for the last two factors in the result.

Proceeding similarly, we arrive at the results in each row of Table 2. Note that the results in some rows (e.g. rows 6, 7, and 8) are negative, and contain factors of $p_-^A$ or $p_-^B$ because they represent the effects of negative feedback that cause modules which are initially above the cortical threshold for both odors to cease to be so.

**Overall equation.** As described in Results, the cortical similarity is given by $\rho = C_{AB}/\sqrt{C_A C_B}$, which is the ratio of the number of modules with cortical responses to both odors to the geometric mean of the numbers of modules with cortical responses to each of $A$ and $B$. Thus the similarity after feedback is:

$$\rho_f = \frac{C_{AB} + \delta C_{AB}^{\text{total}}}{\sqrt{(C_A + \delta C_A^{\text{total}})(C_B + \delta C_B^{\text{total}})}} = Q_3 \frac{N_{AB}}{\sqrt{N_A N_B}} + Q_1 \tag{18}$$

where in the second equation we are separating out all terms proportional to $N_{AB}$, having noting that, from (13), $C_{AB}$ is proportional to $N_{AB}$, and that, from the expressions for $\delta C_{AB}^{\text{total}}$ in Table 2 and for the densities $\rho(AB)$, $\rho(\bar{A}B)$, $\rho(A\bar{B})$, and $\rho(\bar{A}\bar{B})$ in Table 1, every term in $\delta C_{AB}^{\text{total}}$ also contains one piece proportional to $N_{AB}$. Upon including the explicit expressions for $\delta C_A^{\text{total}}$, $\delta C_B^{\text{total}}$, and $\delta C_{AB}^{\text{total}}$ all factors of $N_A$, $N_B$ and $N_{AB}$ except those explicitly displayed in (18) cancel out, so that $Q_3$ and $Q_1$ only depend on the remaining model parameters such as the thresholds and the probabilities of positive and negative feedback. Recalling from (14) that the initial cortical similarity before feedback is $\rho_i = Q_0 N_{AB}/\sqrt{N_A N_B}$, we can express the change in similarity as

$$\Delta\rho = Q_2\,\rho_i + Q_1 \quad \text{with} \quad Q_2 = \frac{Q_3}{Q_0} - 1 \tag{19}$$

This is the main result of our analysis.

**Pattern convergence and divergence: Analytical model.** Fig 3 illustrates these general results for a wide variety of feedback parameters. The general finding is that if the cortical activation threshold is sufficiently high, then $Q_1 = 0$ and $Q_2 > 0$ when the feedback similarity is high, while $Q_2 < 0$ when the feedback similarity is low. Thus high and low feedback similarity give rise to, respectively, pattern convergence and divergence increasing with initial odor similarity. On the contrary, if the cortical threshold is low, so that $\theta_c < \theta_m + \Delta R$, then $Q_1 \neq 0$ and $Q_2 < 0$ for both low and high feedback similarities (partial reversal of effects, Fig 5 and green regions in Fig 3). Here, we discuss in more detail the intercept $Q_1$ and the sign of the slope $Q_2$ in high and low cortical-threshold regimes.

*Intercept $Q_1$ and slope $Q_2$ in the high-threshold regime.* The coefficient $Q_2$ in (19) arises from the terms in $\delta C_{AB}^{\text{total}}$ that are proportional to $N_{AB}$, the number of modules that are responsive to

both odors. Every contribution $\delta C^{(i)}$ to $\delta C_{AB}^{\text{total}}$ that is listed in Table 2 contains such a term via the expressions in Table 1 for the densities of modules that are responsive/unresponsive to the two odors, $\rho(AB)$, $\rho(\bar{A}B)$, $\rho(A\bar{B})$, and $\rho(\bar{A}\bar{B})$. By contrast, the intercept $Q_1$ comes only from the contributions 9–15 in Table 2 that arise from situations where modules are non-responsive before feedback for at least one of the two odors ($R < \theta_m$). These contributions only matter if the effect of feedback, $\Delta R$, is larger than the difference between the cortical and response thresholds, $\Delta\theta$ (hence the step function $H(\Delta R - \Delta\theta)$ in each of these contributions). Thus all of these contributions vanish if the cortical activation threshold is sufficiently high, i.e. $\theta_c > \theta_m + \Delta R$, so that $Q_1 = 0$ in these cases (Fig 2). In Results, we have also provided explanations for the sign of $Q_2$ in the high cortical-threshold regime, in the simple situations of identical ($Q_2 > 0$) and opposite ($Q_2 < 0$) feedback. The only exception to these situations is the case of $\sim 100\%$ inhibitory feedback for both odors (i.e. $p_-/(p_+ + p_-) \sim 1$). In this singular case, identical feedback yields pattern divergence increasing with odor similarity ($Q_2 < 0$). As a simple illustration, consider the special case that the inhibitory feedback targets all the modules ($p_-^A = p_-^B = 1$): the effect of such feedback on the cortical similarity is equivalent to an increase in the cortical activation threshold for both odors, which corresponds to flipping the sign of $\Delta R$ in Eq (8). Thus $Q_2 = -\Delta R/(R_{\max} - \theta_c) < 0$. As another special case, the slope is also negative when feedback does not target all the modules and affects disjoint subsets for the two odors.

*Intercept $Q_1$ and slope $Q_2$ in the low-threshold regime.* Now consider a situation where cortex has increased excitability and thus has a low threshold relative to the strength of feedback: $\theta_c < \theta_m + \Delta R$. Then, from Eq (19) we can derive exact expressions for $Q_1$ and $Q_2$ in the special case $p_+^A = 1$, $p_{\text{both}} = 1$, $p_{\text{flip}} = 0$, i.e., when feedback targets all the modules and is positive for both odors (Fig 5A, top). In this case, $p_+^B = 1$ as well and $p_-^A = p_-^B = 0$. Because of this, only the $p_+^A$ and $p_+^B$ terms in Eqs (15), (16) and (17) contribute to the shifts in the number of cortically responsive modules $\delta C_A^{\text{total}}$ and $\delta C_B^{\text{total}}$ that appear in the denominator of the cortical similarity after feedback (18).

Every contribution to $\delta C_{AB}^{\text{total}} = \sum_{i=1}^{15} \delta C^{(i)}$ in the numerator of the similarity Eq (18) contains a term that is *proportional* to $N_{AB}$ because there is such a term in all module densities $\rho$ listed in Table 1; we will call these terms $\delta C^{(i,p)}$. Meanwhile the contributions $\delta C^{(i)}$ with $i = 9, \cdots 15$ also contain terms that are *not proportional* to $N_{AB}$; we will call these $\delta C^{(i,n)}$. In terms of these quantities, after some algebra we can write the intercept $Q_1$ and slope $Q_2$ in (19) as:

$$
\begin{aligned}
Q_1 &= \frac{\sum_9^{15} \delta C^{(i,n)}}{\sqrt{(C_A + \delta C_A^{\text{total}})(C_B + \delta C_B^{\text{total}})}} \\
Q_2 &= \frac{\sqrt{N_A N_B}}{Q_0 N_{AB}} \frac{(C_{AB} + \sum_1^{15} \delta C^{(i,p)})}{\sqrt{(C_A + \delta C_A^{\text{total}})(C_B + \delta C_B^{\text{total}})}} - 1
\end{aligned}
\tag{20}
$$

Notice that the intercept $Q_1$ only takes contributions from modules that do not respond to at least one odor before feedback (cases $i = 9 \cdots 15$ in Table 2), and thus requires a low cortical threshold with $\Delta\theta < \Delta R$. Explicit computation shows that $Q_1 > 0$. Meanwhile, in the expression for $Q_2$ there are three qualitatively different sets of terms:

- The part of the sum for $Q_2$ from $i = 1$ to $i = 8$ represents modules that initially respond to both odors (see Table 2), and these give positive contributions of $Q_2$, just like they do in the high cortical threshold regime.

- The term in $Q_2$ with $i = 9$ comes from modules that do not respond to either odor before feedback (see Table 2), and therefore only makes a contribution to $Q_2$ in the low threshold

regime where the difference in thresholds is smaller than the effect of feedback, i.e., $\Delta\theta < \Delta R$. This term is positive because the number of modules in this set increases as odor similarity increases.

- The part of the sum in $Q_2$ from $i = 10$ to $i = 15$ arises from modules that respond to only one of the two odors before feedback (see Table 2). Therefore it also only makes a contribution to $Q_2$ when the cortical threshold is low, i.e., $\Delta\theta < \Delta R$. This contribution is negative because the number of modules in this set increases as the odor similarity *decreases*.

Overall, the third contribution dominates so that the slope is negative ($Q_2 < 0$) when $\theta_c \leq \max(\theta_m, \Delta R)$, regardless of the other parameter values. This is a sufficient condition to have $Q_2 < 0$, but for most values of the model parameters, higher $\theta_c$ can also yield negative slopes, as long as $\theta_c < \theta_m + \Delta R$.

A similar argument can be applied to the case $p_+^A = 1$, $p_{\text{both}} = 1$, $p_{\text{flip}} = 1$, i.e. when feedback targets all the modules and is positive for odor A, and negative for odor B (Fig 5A, bottom). In this case the analysis shows that in this feedback condition $Q_2 < 0$ for any $\theta_c < \theta_m + \Delta R$ and $Q_1 > 0$ for any $\theta_c < \min(\theta_m + \Delta R, R_{\max} - \Delta R)$.

### The anatomically detailed network

**Connectivity.**   Each cell was modeled by its distribution of lateral dendrites (for the MCs) or its dendritic spines (for the GCs), since the interface between these two cell types occurs at the junction of these neuronal processes. To determine the average number of synapses between a specific GC and MC then, we wished to calculate the average number of spines from that GC that would be present at a sufficiently close space to the MC lateral dendrites, under the assumption that spines which are sufficiently close to the dendrites will form synapses. Thus, in geometric terms, we wished to calculate, for a given MC-GC pair, 1) the amount of lateral dendrite present in the overlap between the MC lateral dendritic and GC spine spatial distributions and then 2) the number of spines sufficiently close to those lateral dendrites in the overlap.

To treat this mathematically, we first developed mean-field distributions of MC lateral dendritic length density (in $\mu$m of lateral dendrite per unit area) and GC spine density (number of spines per unit volume) based on the experimental literature. Since the MC lateral dendritic distribution was defined on a disk and assumed to be radially symmetric, it was simply dependent on the distance away from the center of the cell, with the specific shape of this distribution determined from data from [70–72]. Meanwhile, the GC spine distribution was defined on an inverted cone. The spine density at a given height was expressed as the number of spines per unit height $N_s$ divided by the area of the circular cross-section of the cone at that height. We assumed no radial dependency, since the dendrites carrying the spines pass roughly perpendicular to the MC dendrites, but $N_s$ was height-dependent: this height dependence was modeled based on results from [70] and [73]. Thus, the MC and GC distributions could be expressed as $\rho_m(r)$ and $\rho_g(z)$, respectively, where $r$ is the distance away from the center of the MC and $z$ is the height along the GC cone (Fig 4A).

With these distributions, we could then calculate the number of expected synapses between a given MC-GC pair. First, we determined how much length of lateral dendrite (in $\mu$m) existed in the overlap between the dendritic distributions of the two cells via the integral:

$$l = \int_{A_{\text{overlap}}} \rho_m(r)dA \tag{21}$$

To calculate the number of spines apposed to these dendrites (and therefore synapses), we needed to account for the 3-dimensional nature of potential synapses between MC and GC by

determining the total volume of MC-GC interaction around these dendrites. To do this, we first converted the total dendritic length in the overlap to an equivalent volume of lateral dendrite by assuming the lateral dendrites to be cylinders with diameter 1 $\mu$m. Then the volume is simply the area of the cross-section multiplied by the total length in the overlap calculated in 21, $\pi r_{\text{dendrite}}^2 \times l$. We then defined the volume of interaction as a cylindrical shell surrounding the lateral dendrite with a thickness $d_{\text{shell}}$ equal to 1.02 $\mu$m, about the effective diameter of a spine [109] (S4A Fig). To calculate this volume, we simply subtracted the lateral dendrite volume from the combined volume of the shell and lateral dendrite as follows:

$$V = \pi((d_{\text{shell}} + r_{\text{dendrite}})^2 - r_{\text{dendrite}}^2)l = q\pi l \tag{22}$$

Where $q = 2.32$ $\mu m^2$ [72]. The density of GC spines in this volume was determined by the height where the MC disk intersects the GC cone, under the simplifying assumption that this density is roughly constant across the height of the volume of interest. Finally, we can multiply this spine density by the previous volume $V$ to find the average number of spines and thus synapses present between MC and GC:

$$\lambda = \rho_g(z_{\text{intersect}})V = \rho_g(z_{\text{intersect}})(q\pi \int_{A_{\text{overlap}}} \rho_m(r)dA) \tag{23}$$

Since a given MC-GC pair generally has only one synapse [109], we used a Poisson distribution to calculate the probability that at least one synapse was formed, with the assumption that if more than one synapse was formed, it would still be counted as a single synapse. Thus:

$$P(\text{synapse}) = P(N_{\text{synapses}} \neq 0) = 1 - \exp(-\lambda) \tag{24}$$

From this algorithm, we were able to generate a network of 3,550 MCs and 53,250 GCs for simulation.

**Neuronal and network dynamics.** The internal dynamics of individual neurons were modeled via the Izhikevich equations [63, 64]:

$$C\frac{dv}{dt} = k(v - v_r)(v - v_t) - u + I \quad ; \quad \frac{du}{dt} = a(b(v - v_r) - u),$$

with spike reset:

$$\text{If } v \geq v_c, \text{then} \begin{cases} v \leftarrow c \\ u \leftarrow u + d \end{cases},$$

where $v$ is the voltage, $u$ is a recovery current, $v_r$ is the resting potential, $v_t$ is the threshold potential, $v_c$ is the cutoff voltage, and $I$ is external current; $a$, $b$, $c$, $d$, and $k$ are parameters to be chosen.

For mitral cells, the parameters were selected to account for their class II behavior [110] and to establish a realistic $f - I$ curve [72]. Granule cells were assumed to be class I neurons; in that case, in the Izhikevich model, $b$ and $k$ could be calculated as follows for $b < 0$ [64]:

$$b = \frac{v_r - v_t + 4R\rho}{4R^2\rho} \quad ; \quad k = \frac{1}{4R^2\rho},$$

where $R$ is input resistance and $\rho$ is the rheobase. The remaining parameters were chosen to match a realistic $f - I$ curve [73]. The mean parameter values are given in Table 3. Parameter values were drawn from a normal distribution with standard deviation equal to 1/10 of the given mean, except for $b$ and $k$ for granule cells, which were drawn from normal distributions

**Table 3. Mean values of the neuronal parameters in the anatomically detailed model.**

| Neuronal parameters (mean values) | | |
|---|---|---|
| Variable | Mitral cell | Granule cell |
| $k$ (nS(mV)$^{-1}$) | 2.5 | 0.067 |
| $a$ (ms$^{-1}$) | 0.02 | 0.01 |
| $b$ (nS) | 12 | -0.133 |
| $c$ (mV) | -70 | -75 |
| $d$ (pA) | 13 | 2 |
| $v_r$ (mV) | -58 | -71 |
| $v_t$ (mV) | -49 | -39 |
| $v_c$ (mV) | 30 | 25 |
| $C$ (pF) | 191 | 48 |

with standard deviations equal to 2/3 of the mean and constrained such that 1) $b < 0$, 2) the resulting rheobase was between 10 and 70 pA, and 3) the input resistance was between 0.25 and 1.5 GΩ.

The dendrodendritic synapses were modeled as NMDA and AMPA receptors on GCs and GABA receptors on MCs. The synaptic current was modeled as the following for AMPA receptors:

$$I_{\text{AMPA}}(t) = s(t)g_{\text{AMPA}}(V(t) - E),$$

where, for a particular receptor, $g$ is the conductance, $s$ is a gating variable representing the fraction of open channels at the synapse, $V(t)$ is the voltage of the recipient cell, and $E_e = 0$ mV is the reversal potential.

For GABA receptors, the current was modeled as:

$$I_{\text{GABA}}(t) = s(t)g_{\text{GABA}}\exp\left(-\frac{L}{\lambda}\right)(V(t) - E_i),$$

where $E_i = -70$ mV is the inhibitory reversal potential, $L$ is the distance between the center of the MC and the synapse location, and $\lambda$ is a length constant. Since inhibitory signals from the cell periphery degrade as they propagate to the soma [111], we modeled this degradation as a simple exponential decay.

Additionally, for NMDA receptors [112]:

$$I_{\text{NMDA}}(t) = s(t)\frac{g(V(t) - E)}{1 + \dfrac{[\text{Mg}^{2+}]\exp(-0.062V(t))}{3.57}},$$

where the additional term describes the magnesium block, with [Mg$^{2+}$] assumed to be 1 mM.

The time evolution of the gating variables for each receptor was modeled following [113]:

$$\frac{ds_{\text{AMPA}}}{dt} = \frac{-s_{\text{AMPA}}}{\tau_{\text{AMPA}}} \quad ; \quad \frac{ds_{\text{GABA}}}{dt} = \frac{-s_{\text{GABA}}}{\tau_{\text{GABA}}}$$

$$\frac{ds_{\text{NMDA}}}{dt} = \frac{-s_{\text{NMDA}}}{\tau_{NMDA_{decay}}} + \alpha n(t)(1 - s_{\text{NMDA}}(t)) \quad ; \quad \frac{dn}{dt} = \frac{-n}{\tau_{NMDA_{rise}}}$$

When a MC spikes, the gating variables of NMDA and AMPA receptors at its synapses (which provide NMDA and AMPA current to the connected GCs) are updated as follows:

$$s_{\text{AMPA}} \leftarrow s_{\text{AMPA}} + W(1 - s_{\text{AMPA}}) \quad ; \quad n \leftarrow n + W(1 - n),$$

where $W = 0.5$.

Additionally, the gating variables of all MCs indirectly connected to the spiking MC via GCs have the gating variables of their GABA receptors at their synapses with those GCs updated as follows to account for spike-independent GC spiking [114]:

$$s_{\text{GABA}} \leftarrow s_{\text{GABA}} + \kappa W(1 - s_{\text{GABA}}),$$

where $0 < \kappa < 1$.

When a GC spikes, the gating variables of GABA receptors at its synapses (providing GABA current to all connected MC) are updated as follows:

$$s_{\text{GABA}} \leftarrow s_{\text{GABA}} + W(1 - s_{\text{GABA}})$$

The values of the time constants $\tau$ and $\alpha$ were taken or derived from [85], while the synaptic conductances and $\kappa$ were tuned in a specialized network, whose dimensions were meant to mimic (within computational constraints), the length and width of a brain slice, in order to reproduce as faithfully as possible lateral inhibition results from [76]. The length constant $\lambda$ was calculated from the formula in [111] for a 1.26 $\mu$m diameter dendrite. The mean parameter values are given in Table 4. Parameter values were drawn from a normal distribution with standard deviation equal to 1/10 of the given mean. Simulations were performed via a forward Euler method with a time step of 0.1 ms.

**Firing rate distributions.** Firing rate distributions were acquired by simulating a network of 3,550 mitral cells and 53,250 granule cells for one sniff cycle ($\frac{1}{6}$ of a second assuming a 6 Hz sniffing rate) for a set of 8 different odors each targeting 20 of 178 glomeruli, with the strength of the odor being drawn from a uniform distribution between 300 and 500 pA. The resulting firing rates were fit with a generalized Pareto distribution with parameters $k = -0.281$, $\sigma = 3.331$, and $\theta = -0.4$ (S4(B) Fig). Excitatory feedback current was randomly added to a different subset of the MCs for each odor (in each case, feedback targeted $\frac{1}{5}$ of all MCs with current drawn from between 0 and 300 pA), and the simulations were rerun. The resulting changes in firing rates due to feedback were fit by a zero-elevated skew normal distribution (S4(C) Fig), with the probability of being drawn from the distribution equal to 0.31 and the consequent

**Table 4. Mean values of the synaptic parameters in the anatomically detailed model.**

| Synaptic parameters (mean values) | |
|---|---|
| **Variable** | **Value** |
| $g_{\text{AMPA}}$ (nS) | 0.73 |
| $g_{\text{NMDA}}$ (nS) | 0.84 |
| $g_{\text{GABA}}$ (nS) | 0.13 |
| $\kappa$ | 0.006 |
| $\tau_{\text{AMPA}}$ (ms) | 5.5 |
| $\tau_{\text{NMDA}_{\text{rise}}}$ (ms) | 10 |
| $\tau_{\text{NMDA}_{\text{decay}}}$ (ms) | 80 |
| $\tau_{\text{GABA}}$ (ms) | 18 |
| $\lambda$ ($\mu$m) | 675 |
| $\alpha$ (ms$^{-1}$) | 0.1 |

skew normal distribution having the parameters $\alpha$ = 4.232, $\omega$ = 2.687, and $\xi$ = 0.7785 (S4(C) Fig, inset). For excitatory feedback to GCs, a simulation was done for a set of 8 odors each targeting 20 glomeruli, with a different feedback pattern for each odor (targeting $\frac{1}{8}$ of all GCs with current drawn from between 0 and 200 pA); the change in firing among the odor-receiving cells was then determined. Note that the distribution was determined among MCs with spike count greater than or equal 2, as the maximum number of spikes lost due to GC feedback was found to be 2. The resulting changes in firing rate were fit with a lognormal distribution with parameters $\mu$ = 0.7957, $\sigma$ = 0.2548, which was then appropriately shifted and flipped to produce values with the desired range and shape (S4(D) Fig).

## Supporting information

**S1 Fig. The trends in pattern convergence and divergence generalize to realistic normal distributions of module responses to odor and feedback inputs.** (**A**) Results obtained from the statistical model with mean and variance of the normal distributions: $\mu_1$ = 1.15, $\sigma_1^2 = 0.42$ for odor inputs, and $\mu_2$ = 0.57, $\sigma_2^2 = 0.28$ for feedback inputs. The cortical threshold is set to $\theta_c$ = 1.6 to give cortical activation of 10% [25]. Datapoint obtained by averaging the results for 10 randomly generated pairs of odor inputs with close similarity values. Different markers indicate different feedback conditions. Filled red/blue circles: the two feedback scenarios of Fig 2, corresponding to $\rho_{FB}$ = 0.8 and $\rho_{FB}$ = −0.8, respectively. Note that $|\rho_{FB} \neq 1|$ due to variability in the amplitude of the feedback strength $\Delta R$. Empty red/blue circles: two intermediate conditions, corresponding to 50% of modules receiving feedback, of which 75% are shared between the two odors. Of the feedback-targeted modules, 75% receive inhibitory feedback for both odors in the first case (red, $\rho_{FB}$ = 0.6), and 75% / 12.5% receive inhibitory feedback for the first/second odor in the second case (blue, $\rho_{FB}$ = −0.6) (**B–E**) The parameters of the distributions are varied one by one with respect to (A). Results are robust against changes in (**B**, **C**) the mean and variance, respectively, of the distributions of module responses to odor inputs, and (**D**, **E**) the mean and variance, respectively, of the distributions of module responses to feedback inputs. As in (A), each datapoint is obtained by averaging the results for 10 randomly generated pairs of odor inputs with close similarity values. Different markers indicate different feedback conditions. Filled red/blue circles: the two feedback scenarios of Fig 2, corresponding to $\rho_{FB}$ = 0.8 and $\rho_{FB}$ = −0.8 in (**B–C**), $\rho_{FB}$ = 0.75 and $\rho_{FB}$ = −0.75 in (**D, left**), $\rho_{FB}$ = 0.94 and $\rho_{FB}$ = −0.94 in (**D, right**), $\rho_{FB}$ = 0.92 and $\rho_{FB}$ = −0.92 in (**E, left**), $\rho_{FB}$ = 0.72 and $\rho_{FB}$ = −0.72 in (**E, right**). Empty red/blue circles: two intermediate conditions, corresponding to 50% of modules receiving feedback, of which 75% are shared between the two odors. Of the feedback-targeted modules, 75% receive inhibitory feedback for both odors in the first case (red, $\rho_{FB}$ = 0.6 in (**B–C**), $\rho_{FB}$ = 0.56 in (**D, left**), $\rho_{FB}$ = 0.71 in (**D, right**), $\rho_{FB}$ = 0.69 in (**E, left**), $\rho_{FB}$ = 0.54 in (**E, right**)), and 75% / 12.5% receive inhibitory feedback for the first/second odor in the second case (blue, $\rho_{FB}$ = −0.6 in (**B–C**), $\rho_{FB}$ = −0.56 in (**D, left**), $\rho_{FB}$ = −0.71 in (**D, right**), $\rho_{FB}$ = −0.69 in (**E, left**), $\rho_{FB}$ = −0.54 in (**E, right**)).
(TIF)

**S2 Fig. The mechanism for pattern convergence and divergence requires a network with (at least) two layers linked by a high-threshold transfer function.** The statistical effects induced by feedback in a single-layer architecture or with low cortical threshold are qualitatively different from those arising in a two-layer model with a high-threshold transfer function. (**A**) With only one layer, pattern convergence can be achieved only when the feedback similarity is very high and decreases with increasing odor similarity (red). Moderately correlated and anticorrelated feedback induce similar effects (empty red and blue circles). Same feedback conditions as in (C): Filled red/blue circles for the two extreme feedback scenarios with $\rho_{FB}$ = 0.8

and $\rho_{FB}$ = −0.8, respectively; empty red/blue circles for the two intermediate feedback conditions, with $\rho_{FB}$ = 0.6 and $\rho_{FB}$ = −0.6, respectively. (**B**) Same conditions as in (C) except with lower cortical threshold. The trend reversal is similar to that seen in the analytical framework (Fig 5A). (**C**) S1A Fig demonstrating results from a normal two-layer, high-threshold architecture for comparison.
(TIF)

**S3 Fig. Pattern convergence that increases with initial odor similarity only occurs for sufficiently high numbers of mitral and cortical cells.** A robustly positive slope of the relationship between initial similarity and change in similarity at high threshold is only achieved for a sufficient number of MCs and bulb modules (i.e., cortical cells). For different numbers of $M$ mitral cells and $K$ cortical cells, we simulated presentation of the same positive feedback for different pairs of odors and then measured the slope of the relationship between initial similarity and change in similarity at high threshold. Although positive slope was achieved for relatively low numbers of MCs and cortical cells, this relationship did not achieve a consistently high $r^2$ value without approximate values of $M > 8000$ and $K > 80000$. For all simulations, $f_{odor}$ = 0.12, $p_{FB}$ = 0.08 (fraction of feedback-targeted MCs), and $q$ = 0.07.
(TIF)

**S4 Fig. Details of the biophysical model and subsequent distributions in firing rates.** (**A**) Depiction of the relevant volumes for calculating MC-GC connectivity (**B**) Distribution of MC firing rates due to odor input (**C**) Distribution of changes in MC firing rates due to excitatory feedback. Inset shows the skew normal distribution that was sampled if the change was not equal to 0 (see "Firing rate distributions") (**D**) Distribution of changes in odor-receiving MC firing rates due to excitatory feedback to GCs. Note that the lognormal distribution has been shifted to match the range of the data. The parameters of the simulations for the distributions in **B–D** are provided in the text of the Methods ("Neuronal and network dynamics; Firing rate distributions").
(TIF)

**S5 Fig. Inhibitory feedback induces pattern divergence in the mechanistic model at high threshold.** For high cortical thresholds, pattern divergence occurred when excitatory feedback was presented for one odor and inhibitory for the other (purple line), or when highly correlated inhibitory feedback was presented for both odors (green line). In all cases, we simulated 10,000 MCs grouped into 500 glomeruli and 100,000 cortical cells, each sampling 7% of the MCs, with odor targeting 12% of the glomeruli, positive feedback targeting 8% of the MCs, and negative feedback targeting all MCs.
(TIF)

**S6 Fig. Sufficiently similar feedback is required to produce a positive relationship between initial odor similarity and change in similarity following excitatory feedback to MCs.** Sufficiently correlated positive feedback produces a proportional relationship between initial similarity and change in similarity. We generated a range of feedback similarities for pairs of excitatory feedback vectors. We found that the slope of the relationship between initial similarity and change in similarity at high threshold varied linearly with the feedback similarity (slope = 0.5469, $r^2$ = 0.9322). Thus, for feedback vectors with significant similarity, the relationship between initial similarity of two odor representations and the change in similarity of those representations following feedback is positive. For all simulations, $M$ = 10000, $K$ = 100000, $f_{odor}$ = 0.12, $p_{FB}$ = 0.08, $G$ = 500, and $q$ = 0.07.
(TIF)

**S1 Text. Pattern convergence and divergence for normal distributions of module responses: Numerical results.**
(PDF)

**S2 Text. The necessity of a nonlinear transfer function with high cortical threshold in simulations of the statistical model.**
(PDF)

## Acknowledgments

We thank Jay A. Gottfried for fruitful discussions. DK would also like to thank Minghong Ma and Graeme Lowe for their advice on olfactory bulb anatomy, Eugenio Piasini for use of his server, as well as Max Kelz, Joseph Aicher, Rebecca Li, and Matthew Kersen for useful discussions and feedback.

## Author Contributions

**Conceptualization:** Gaia Tavoni.

**Formal analysis:** Gaia Tavoni, David E. Chen Kersen.

**Funding acquisition:** Gaia Tavoni, Vijay Balasubramanian.

**Investigation:** Gaia Tavoni, David E. Chen Kersen.

**Methodology:** Gaia Tavoni, David E. Chen Kersen.

**Project administration:** Vijay Balasubramanian.

**Resources:** Vijay Balasubramanian.

**Software:** Gaia Tavoni, David E. Chen Kersen.

**Supervision:** Vijay Balasubramanian.

**Validation:** Gaia Tavoni, David E. Chen Kersen.

**Visualization:** Gaia Tavoni, David E. Chen Kersen.

**Writing – original draft:** Gaia Tavoni, David E. Chen Kersen, Vijay Balasubramanian.

**Writing – review & editing:** Gaia Tavoni, David E. Chen Kersen, Vijay Balasubramanian.

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
