## [Decision Letter · Decision Letter 0]

8 Sep 2021

Dear Dr. Tavoni,

Thank you very much for submitting your manuscript "Cortical feedback and gating in odor discrimination and generalization" for consideration at PLOS Computational Biology. As with all papers reviewed by the journal, your manuscript was reviewed by members of the editorial board and by several independent reviewers. The reviewers appreciated the attention to an important topic. Based on the reviews, we are likely to accept this manuscript for publication, providing that you modify the manuscript according to the review recommendations.

Sincerely,

Alexandre V. Morozov, Ph.D.

Associate Editor

PLOS Computational Biology

Samuel Gershman

Deputy Editor

PLOS Computational Biology

[LINK]

Reviewer's Responses to Questions

**Comments to the Authors:**

Reviewer #1: I really like the article, congratulations to the authors, it proposes a new vision of olfactory perception.

I would like to suggest in discussion to enlarge even only hypothetically:

i. the interactions between the various odorants in the formation of the scent;

ii. the cross-modal interactions for example with the gustatory system;

and finally an important aspect since the ambition of the authors concerns also Alzheimer d. is iii. to discuss how the model can interpret aging and especially olfactory phenotypes (see Oncotarget).

minor points:

- I recommend a re-reading of the paper for minor adjustments, e.g. Ref # 84.

Reviewer #2: Overview

Within this work, the authors develop a model to predict convergence and divergence of cortical patterns in odor perception. This model contributes to understanding the effect of cortical feedback in odor perception.

The authors made a great effort in explaining the concept behind their model and the anatomy it follows. The result is a very well-written and straightforward manuscript.

As a general note, the authors should explain in more detail their hypothesis. I would also expect a more detailed explanation of the reason why they came up with the need to develop the model, what it can be useful for, and what it may bring to the field.

Specific points follow:

Line 77: You state that you have only minimal assumptions about statistics of odor and feedback input. Which are these assumptions? Why do you imply them? You explicit all the assumptions in the method section, but I would expect to explicit which kind of assumptions are you implying, without analyzing them in depth, as it has been done in the methods.

Line 82: explain more in detail what do you mean by semantic context. This is the first time, and also the last one, you refer to a semantic context.

Line 88: the authors mention Alzheimer’s disease as a neurological condition that induces differences in cortical excitability. I would deepen the usefulness of your model from a clinical point of view.

Line 93: It is not clear why you refer to an “initial” model

Line 108: “context as the effect produced by cortical feedback” As you explicit in the abstract, I would also include in the introduction what do you mean by context.

Line 391: the authors state that in the presence of different odors and similar contexts, they would expect greater convergence. Please discuss more the initial hypothesis and the results,

Line 403: your work focused on the olfactory system. Why did you decide to focus on this particular perception and not on other systems?

Line 487: how would you test your predictions in patients with AD?

Line 522: “(i)” instead of “(ii)”

**Have the authors made all data and (if applicable) computational code underlying the findings in their manuscript fully available?**

Reviewer #1: Yes

Reviewer #2: Yes

PLOS authors have the option to publish the peer review history of their article (what does this mean?). If published, this will include your full peer review and any attached files.

Reviewer #1: **Yes: **Andrea Mazzatenta

Reviewer #2: No

Figure Files:

Data Requirements:

Reproducibility:

References:

---

## [Editor Report · Decision Letter 1]

24 Sep 2021

Dear Dr. Tavoni,

We are pleased to inform you that your manuscript 'Cortical feedback and gating in odor discrimination and generalization' has been provisionally accepted for publication in PLOS Computational Biology.

Best regards,

Alexandre V. Morozov, Ph.D.

Associate Editor

PLOS Computational Biology

Samuel Gershman

Deputy Editor

PLOS Computational Biology

---

## [Editor Report · Acceptance letter]

6 Oct 2021

PCOMPBIOL-D-21-01239R1 

Cortical feedback and gating in odor discrimination and generalization

Dear Dr Tavoni,

I am pleased to inform you that your manuscript has been formally accepted for publication in PLOS Computational Biology. Your manuscript is now with our production department and you will be notified of the publication date in due course.

With kind regards,

Agnes Pap
